# Neurexin directs partner-specific synaptic connectivity in *C. elegans*

Alison Philbrook[1], Shankar Ramachandran[1], Christopher M Lambert[1], Devyn Oliver[1], Jeremy Florman[1], Mark J Alkema[1], Michele Lemons[1,2], Michael M Francis[1]*

[1]Department of Neurobiology, University of Massachusetts Medical School, Worcester, United States; [2]Department of Natural Sciences, Assumption College, Worcester, United States

**Abstract** In neural circuits, individual neurons often make projections onto multiple postsynaptic partners. Here, we investigate molecular mechanisms by which these divergent connections are generated, using dyadic synapses in *C. elegans* as a model. We report that *C. elegans nrx-1/* neurexin directs divergent connectivity through differential actions at synapses with partnering neurons and muscles. We show that cholinergic outputs onto neurons are, unexpectedly, located at previously undefined spine-like protrusions from GABAergic dendrites. Both these spine-like features and cholinergic receptor clustering are strikingly disrupted in the absence of *nrx-1*. Excitatory transmission onto GABAergic neurons, but not neuromuscular transmission, is also disrupted. Our data indicate that NRX-1 located at presynaptic sites specifically directs postsynaptic development in GABAergic neurons. Our findings provide evidence that individual neurons can direct differential patterns of connectivity with their post-synaptic partners through partner-specific utilization of synaptic organizers, offering a novel view into molecular control of divergent connectivity.

DOI: https://doi.org/10.7554/eLife.35692.001

*For correspondence: michael.francis@umassmed.edu

Competing interests: The authors declare that no competing interests exist.

## Introduction

Neurons are typically wired into discrete circuits through stereotyped patterns of synaptic connections geared to perform specific functions. Individual neurons within circuits may receive convergent synaptic inputs from multiple classes of presynaptic partnering neurons, and likewise, make divergent synaptic outputs onto distinct postsynaptic targets. We have gained an understanding of some of the core mechanisms that sculpt convergent connectivity through studies of developmental processes such as activity-dependent synapse elimination (*Brown et al., 1976*; *Campbell and Shatz, 1992*; *Okawa et al., 2014a*; *Sanes and Lichtman, 1999*; *Shatz and Kirkwood, 1984*; *Walsh and Lichtman, 2003*). In contrast, the molecular processes controlling the establishment of divergent synaptic connections (between a single presynaptic partner and multiple postsynaptic target cells) are not clearly defined (*Okawa et al., 2014b*). Neural circuit models often represent divergent connections as a means for enabling the same signal from an individual presynaptic neuron to reach many different postsynaptic target cells. However, the strength of connections with postsynaptic partners can vary widely, strongly suggesting that presynaptic neurons have the capacity to establish and regulate connections with each postsynaptic target independently. While molecular guidance cues directing axon outgrowth have been well-studied, an understanding of the molecular mechanisms responsible for directing target-specific connectivity has remained elusive.

A primary mechanism for establishing nascent synapses is through the actions of synaptic adhesion molecules, also known as synaptic organizers (*de Wit and Ghosh, 2016*; *Missler et al., 2012*). These organizers are often anchored to the pre- and post-synaptic membranes (e.g. neurexins,

**eLife digest** Nervous systems are complex networks of interconnected cells called neurons. These networks vary in size from a few hundred cells in worms, to tens of billions in the human brain. Within these networks, each individual neuron forms connections – called synapses – with many others. But these partner neurons are not necessarily alike. In fact, they may be different cell types. How neurons form distinct connections with different partner cells remains unclear.

Part of the answer may lie in specialized proteins called cell adhesion molecules. These proteins occur on the cell surface and enable neurons to recognize one another. This helps ensure that the cells form appropriate connections via synapses. Cell adhesion molecules are therefore also known as synaptic organizers. Philbrook et al. have now examined the role of synaptic organizers in wiring up the nervous system of the nematode worm and model organism *Caenorhabditis elegans*.

Motor neurons form connections with two types of partner cell: muscle cells and neurons. Philbrook et al. screened *C. elegans* that have mutations in genes encoding various synaptic organizers. This revealed that a protein called neurexin must be present for motor neurons to form synapses with other neurons. By contrast, neurexin is not required for the same neurons to establish synapses with muscles. Philbrook et al. found that neuron-to-neuron synapses arise at specialized finger-like projections. These resemble the dendritic spines at which synapses form in the brains of mammals, and had not been previously identified in *C. elegans*. In worms that lack neurexin, these spine-like structures do not form correctly, disrupting the formation of neuron-to-neuron connections.

Previous work has implicated neurexin in synapse formation in the mammalian brain. But this is the first study to reveal a role for neurexin in establishing partner-specific synaptic connections. Mutations in synaptic organizers, including neurexin, contribute to disorders of brain development. These include schizophrenia and autism spectrum disorders. Learning more about how neurexin helps establish specific synaptic connections may help us understand how these disorders arise.
DOI: https://doi.org/10.7554/eLife.35692.002

neuroligins, leucine-rich repeat transmembrane proteins/LRRTMs) and promote synapse formation through trans-synaptic adhesion and signaling. The importance of these processes in establishing proper neural circuit connectivity is highlighted by the links between mutations in genes encoding these synaptic adhesion/organizing molecules and neuropsychiatric and neurodevelopmental disorders, such as autism spectrum disorder and schizophrenia (*Kim et al., 2008*; *Reichelt et al., 2012*; *Rujescu et al., 2009*). Intriguingly, synaptic organizers are capable of acting in a cell-specific manner to promote synapse formation (*Chen et al., 2017*; *Siddiqui et al., 2013*; *Zhang et al., 2015*). Thus, an exciting possibility is that individual neurons could encode connections with alternate synaptic partners through differential deployment of synaptic organizers.

We have investigated this possibility in the motor circuit of the nematode *Caenorhabditis elegans* where individual excitatory cholinergic motor neurons form synapses with both body wall muscles and GABAergic motor neurons. Through a screen for genes that govern the formation of these divergent synaptic connections, we demonstrate that the synaptic organizer *nrx-1*/neurexin directs the outgrowth of previously uncharacterized dendritic spine-like structures and the formation of synaptic connections with GABAergic neurons, but is not required for synaptic connectivity with muscles. Conversely, genes previously shown to be required for cholinergic connectivity with muscles (*Francis et al., 2005*; *Gally et al., 2004*; *Pinan-Lucarré et al., 2014*) are not required for the formation of synapses onto GABAergic neurons. Our findings demonstrate that cholinergic neurons utilize distinct molecular signals to establish synapses with GABAergic motor neurons versus body wall muscles, thus revealing that a single presynaptic neuron establishes divergent connections by employing parallel molecular strategies for the formation of connections with each postsynaptic partner.

## Results

### Clusters of the GFP-tagged acetylcholine receptor subunit ACR-12 are localized to spine-like dendritic protrusions on DD GABAergic neurons

To establish a system to investigate mechanisms instructing synaptic connectivity, we labeled postsynaptic specializations on dorsally directed GABAergic DD neurons using cell-specific expression (*flp-13* promoter) of the GFP-tagged acetylcholine receptor subunit ACR-12. Prior work showed that ACR-12 receptors in GABAergic motor neurons are clustered opposite cholinergic terminals and GABAergic expression of ACR-12::GFP rescues *acr-12* mutant phenotypes (*Barbagallo et al., 2017*; *Petrash et al., 2013*). Moreover, postsynaptic ACR-12::GFP clusters relocate appropriately during developmental synaptic remodeling of the DD neurons, suggesting these clusters faithfully report synaptic inputs (*He et al., 2015*; *Howell et al., 2015*).

The morphology of DD neurons is highly polarized, facilitating clear visualization of the axonal and dendritic neuronal compartments. In the present work, we focus much of our analysis on the spatially isolated neurites of the DD1 neuron (*Figure 1A–C*). In adults, the anterior DD1 process extends from the soma to enter the ventral nerve cord fascicle (the dendritic compartment), where prior EM studies show that approximately 26 synaptic inputs from cholinergic neurons are concentrated (the synaptic region, *Figure 1C,D*) (*White et al., 1978*, *1976*). The process then crosses the longitudinal midline of the worm via a commissural connection and enters the dorsal nerve cord where it forms *en passant* synaptic outputs onto body wall muscles (the axonal compartment) (*Figure 1A,B*) (*White et al., 1976*). We find that ACR-12::GFP receptor clusters in DD1 are confined to the synaptic region of the ventral dendritic process in the mature animal (*Figure 1C*). As *C. elegans* synapses are formed *en passant*, pre- and post-synaptic specializations typically appear, at the light level, to be localized along the main shafts of neuronal processes. Surprisingly, we noted that the majority of ACR-12::GFP clusters do not appear localized to the shaft of the primary DD1 dendritic process, instead appearing to protrude from the primary DD1 dendrite shaft (*Figure 1C*). To investigate this finding in more detail, we examined morphological features in the synaptic region of the DD1 dendrite (*Figure 1D*). Intriguingly, we noted finger-like structures (~0.3–1 μm in length) projecting outward from the DD1 dendrite in this region (*Figure 1D*, *Figure 1—figure supplement 1A*). In contrast, these structures are not present in the asynaptic region of the process immediately adjacent to the cell soma (*Figure 1D*). These protrusions are obscured by the processes of other ventral cord neurons when using promoters that provide for more broad expression (e.g. *unc-47*), and are therefore most clearly identifiable with specific labeling of DD neurons (*Figure 1—figure supplement 1B*). Spine-like protrusions are also clearly identifiable in the dendrites of more posterior DD neurons (*Figure 1—figure supplement 1C*), but are not apparent in a related class of postembryonic born, ventrally directed GABAergic (VD) neurons (*Figure 1—figure supplement 1D*), although the density of VD processes may complicate their detection. The dendritic protrusions concentrate clusters of ACR-12 receptors at their tips (*Figure 1E*), and over 60% of ACR-12::GFP clusters appear localized to protrusions (*Figure 1F*). The dendritic protrusions increase in abundance through larval development, and this increase correlates well with a similar developmental increase in ACR-12::GFP receptor clusters (*Figure 1—figure supplement 2*). Together, our results provide evidence that cholinergic receptors cluster at morphologically distinct finger-like structures present on DD neuron dendrites, raising the interesting possibility that these structures serve similar roles to dendritic spines in the mammalian nervous system.

### Heteromeric ACR-12-containing AChRs are located on the cell surface opposite cholinergic release sites

To explore the above possibility further, we evaluated the spatial relationship between ACR-12 clusters located on these spine-like structures and cholinergic release sites. We found that dendritic protrusions and ACR-12 receptor clusters are both located opposite clusters of cholinergic synaptic vesicles (*Figure 2A,B*), indicating that these likely represent mature synapses. We therefore next investigated whether these clusters indicate post-synaptic receptors residing at the cell surface. To address this question, we inserted an HA epitope tag into the extracellular C-terminus of ACR-12::GFP (diagram in *Figure 2C*) (*Gottschalk and Schafer, 2006*). Injection of Alexa594 conjugated anti-HA antibody into live transgenic animals expressing this construct produces specific labeling in the

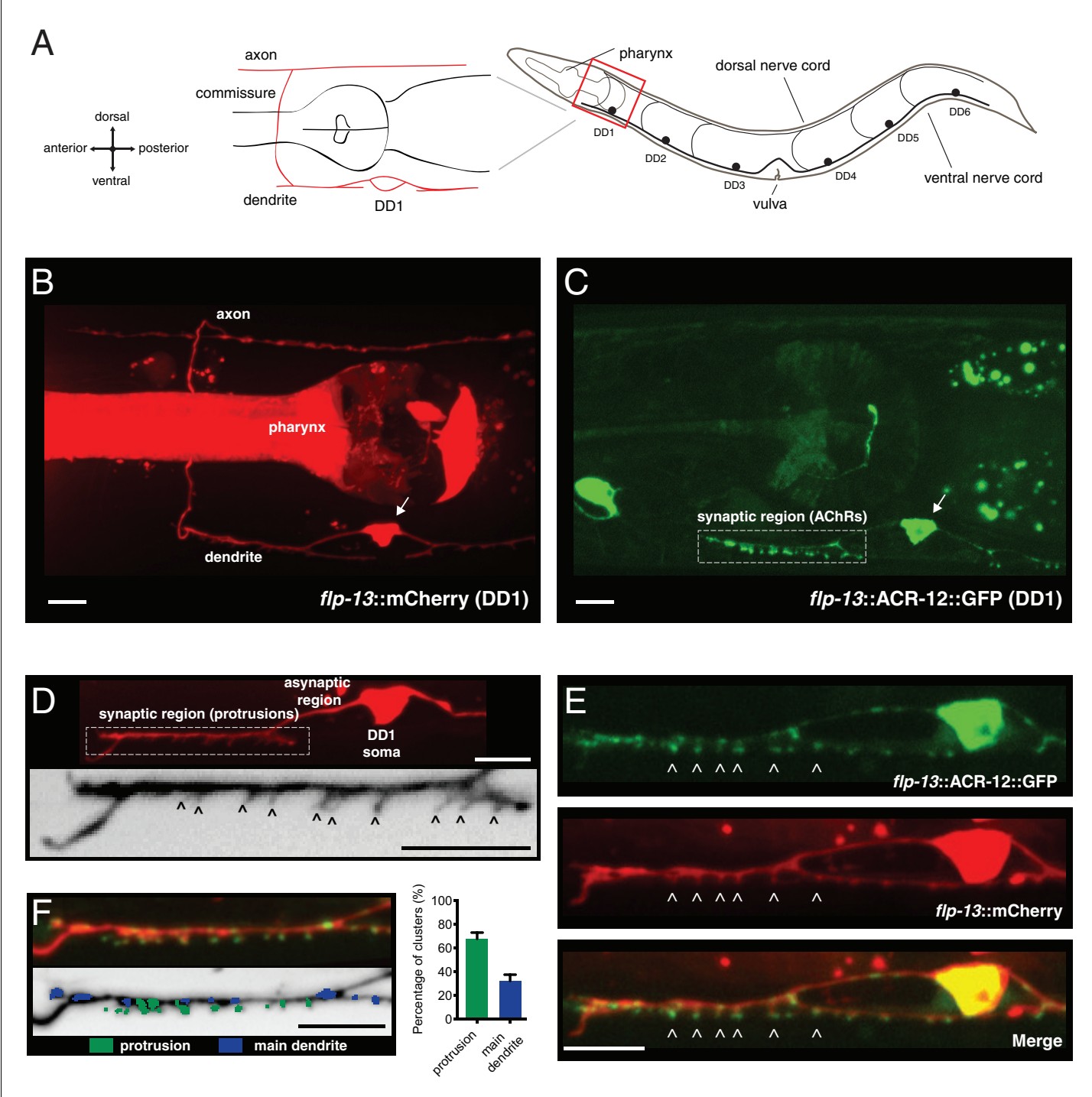

**Figure 1.** Characterization of post-synaptic specializations in the DD neurons. (A) Diagrams of *C. elegans* showing the six DD GABAergic neurons in the ventral nerve cord and expanded view of DD1. After the L1/L2 transition, the DD neurons exclusively make dorsal synaptic outputs onto muscles (axon), while receiving a majority of synaptic inputs on the ventral side (dendrite). (B) Morphology of the DD1 neuron, visualized by expression of the *flp-13*::mCherry transcriptional reporter. Pharyngeal fluorescence indicates expression of the *lgc-11*::mCherry co-injection marker. Arrow indicates DD1 cell body. For this and all subsequent figures, images of L4 animals are shown unless otherwise noted. (C) Cholinergic ACR-12 receptors (*flp-13*::ACR-12::GFP) are localized to a defined region of the DD1 dendritic compartment, labeled as the synaptic region (boxed). Arrow indicates DD1 cell body. (D) Top, confocal image of the DD1 dendritic region visualized by expression of *flp-13*::mCherry. Bottom, inverted image showing expanded view of the synaptic region containing dendritic protrusions (indicated by arrowheads). (E) Confocal images of DD1 soma and synaptic region with coexpression of *flp-13*::ACR-12::GFP and *flp-13*::mCherry. Arrowheads indicate ACR-12 clusters located at the tips of dendritic protrusions. (F) Left, representative confocal image showing the distribution of ACR-12::GFP clusters. ACR-12::GFP receptor clusters associated with either protrusions (green) or the main

*Figure 1 continued on next page*

*Figure 1 continued*

dendritic shaft (blue) are indicated. Right, the percentage of clusters classified into each category (142 receptor clusters from 11 animals were analyzed). Scale bars, 5 µm (**B–F**).

DOI: https://doi.org/10.7554/eLife.35692.003

The following source data and figure supplements are available for figure 1:

**Source data 1.** Raw values for synaptic vs asynaptic ACR-12::GFP receptor clusters.
DOI: https://doi.org/10.7554/eLife.35692.006
**Source data 2.** Raw values for wild type spine length, numbers of spines and ACR-12 receptor clusters throughout development.
DOI: https://doi.org/10.7554/eLife.35692.007
**Figure supplement 1.** Spine-like protrusions are located on DD dendrites.
DOI: https://doi.org/10.7554/eLife.35692.004
**Figure supplement 2.** Spine-like protrusions increase developmentally in a correlated manner with ACR-12 receptor clusters.
DOI: https://doi.org/10.7554/eLife.35692.005

DD1 synaptic region, and is also evident in coelomocytes (scavenger cells that take up excess antibody), confirming successful injection (*Figure 2C*). The anti-HA signal colocalizes with ACR-12::GFP clusters in the synaptic region of DD1, but is not evident in the cell soma. In contrast, the intracellular GFP moiety produces fluorescence that is evident in both the soma and the synaptic region of the dendrite, representing both synaptic and internal receptor pools (*Figure 2C*). Injection of anti-GFP antibody did not produce specific labeling, confirming that the intracellularly positioned GFP is not accessible to antibody (*Figure 2—figure supplement 1A*). Our analysis of ACR-12 localization in DD1 indicates that ACR-12 is incorporated into mature receptor complexes that are specifically targeted for transport to DD neuron dendrites, and reside on the cell surface at post-synaptic sites.

We next sought to gain an understanding of the subunit composition of ACR-12 receptors in GABAergic neurons. Most acetylcholine receptors are formed as heteromeric combinations of five subunits. Prior work has demonstrated that partially or improperly assembled acetylcholine receptor intermediates are not transported out of the ER and are instead targeted for degradation (*Blount and Merlie, 1990*; *Merlie and Lindstrom, 1983*). We therefore reasoned that genetic ablation of obligate ACR-12 partnering subunits, by interfering with assembly, transport, and synaptic targeting of ACR-12 receptor complexes, could provide an efficient strategy for identifying subunit partners. We found that single mutations in the acetylcholine receptor subunit genes *unc-38, unc-63, lev-1* or *unc-29* strongly decrease ACR-12::GFP clustering. In wild type, we observe approximately 15 receptor clusters within the DD1 synaptic region. These clusters are eliminated almost completely with mutation of these subunit genes (*Figure 2—figure supplement 1B,C*). In contrast, mutations in other nAChR subunit genes with previously reported neuronal expression, such as *acr-9* and *acr-14* (*Cinar et al., 2005*; *Fox et al., 2005*), do not disrupt synaptic ACR-12 clustering (*Figure 2—figure supplement 1B*). Mutations in genes important for AChR assembly and trafficking (*unc-50, unc-74*, and *ric-3*) (*Boulin et al., 2008*; *Eimer et al., 2007*; *Halevi et al., 2002*; *Haugstetter et al., 2005*) also abolish synaptic clusters of ACR-12::GFP (*Figure 2—figure supplement 1B,C*). GABA neuron-specific expression of wild type cDNAs encoding individual AChR subunits (UNC-63 or UNC-38) or accessory proteins in the respective mutants is sufficient to restore ACR-12 clustering to wild type levels, providing support that both gene classes act cell-autonomously in GABA neurons to promote receptor assembly, maturation and synaptic delivery (*Figure 2—figure supplement 1B,C*). Cell-specific expression of either UNC-29::GFP or UNC-63::GFP in DD neurons produces punctate labeling in the DD1 synaptic region that closely resembles ACR-12::GFP clustering (*Figure 2D*). Mutation of *acr-12* in these transgenic animals significantly reduces UNC-29::GFP or UNC-63::GFP receptor clusters and fluorescence signal in the DD1 synaptic region (*Figure 2D*, *Figure 2—figure supplement 1D,E*), providing further evidence that they coassemble with ACR-12. Together, our results indicate UNC-38, UNC-63, LEV-1, UNC-29 and ACR-12 subunits coassemble into a pentameric acetylcholine receptor in GABAergic neurons.

## Neurexin directs cholinergic connectivity with GABAergic neurons

To decipher molecular mechanisms by which these newly defined post-synaptic structures develop, we examined ACR-12::GFP labeling in DD neurons of 28 strains carrying mutations in 36 candidate genes. These candidates predominantly encode scaffold and cell-cell interaction proteins previously

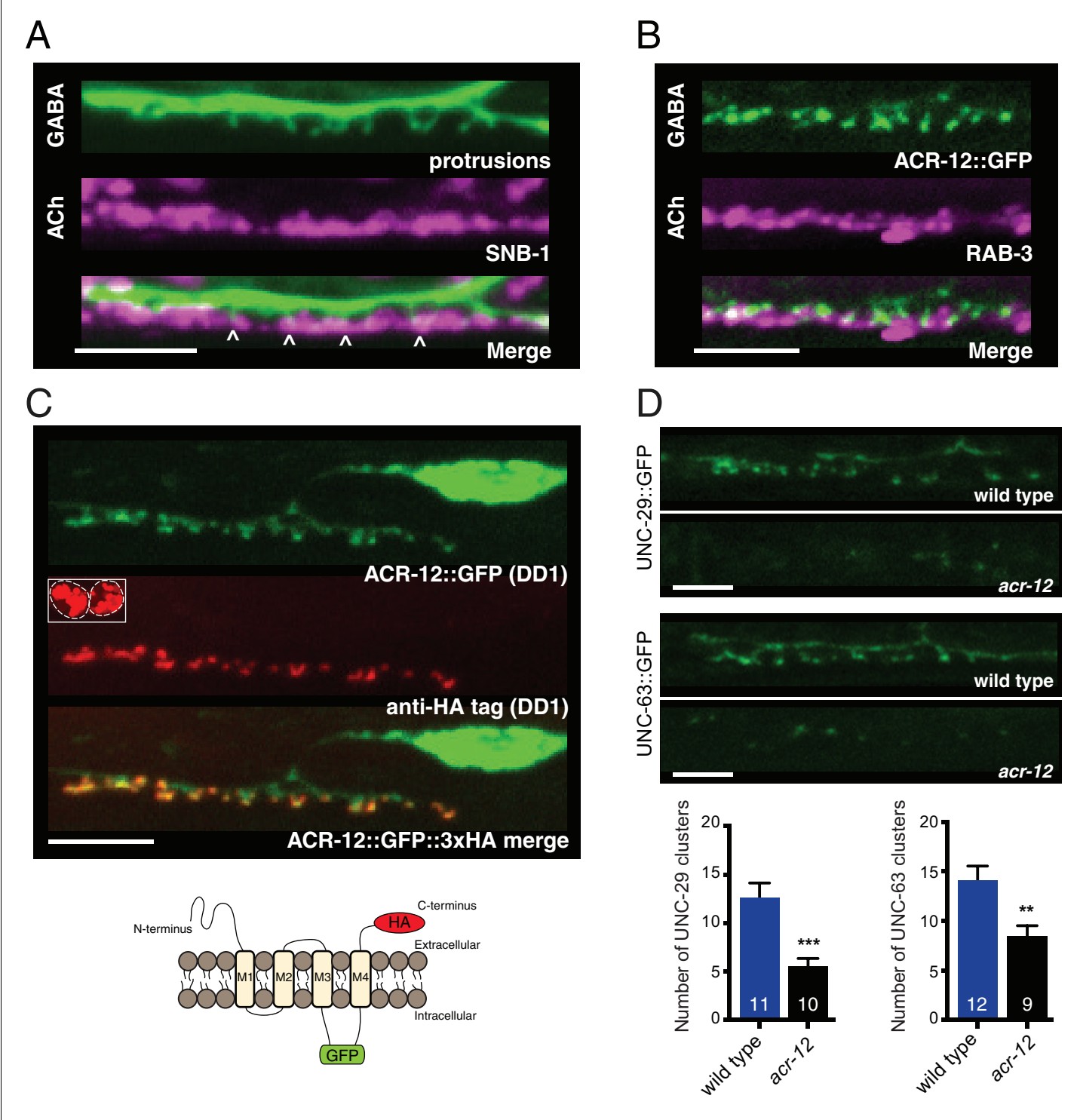

**Figure 2.** Heteromeric ACR-12 receptors are localized at the cell surface opposite sites of release. (A) Confocal images of presynaptic (*acr-2*::SNB-1:: GFP) and postsynaptic (*flp-13*::mCherry) domains in the DD1 synaptic region. Note violet/green coloring to more clearly depict presynaptic structures and protrusions (arrowheads). (B) Confocal images showing apposition of pre- and post-synaptic components with coexpression of the cholinergic synaptic vesicle marker *acr-2*::mCherry::RAB-3 (violet) and the AChR reporter ACR-12::GFP (green) in the DD1 synaptic region. (C) Top, confocal images showing ACR-12 receptor clusters as visualized by GFP fluorescence (green) or anti-HA antibody fluorescence (red) 6 hr following antibody injection. Note the extracellular location of the HA epitope tag (schematic below), enabling selective visualization of synaptic receptor clusters at the cell surface. Inset, anti-HA uptake by coelomocytes indicating successful injection. (D) Top, confocal images of UNC-29::GFP and UNC-63::GFP clusters in the DD1 dendrite (*flp-13* promoter) in wild type or *acr-12(ok367)* mutants. Bottom, quantification of the average number of UNC-29::GFP and UNC-63::GFP

*Figure 2 continued on next page*

*Figure 2 continued*

clusters in the DD1 dendrite for wild type and *acr-12(ok367)* mutants. Each bar represents the mean ± SEM. For this and all subsequent figures, numbers in bars indicate the n for each genotype. **p<0.01, ***p<0.001, student's t-test. Scale bars, 5 μm (**A–D**).

DOI: https://doi.org/10.7554/eLife.35692.008

The following source data and figure supplement are available for figure 2:

**Source data 1.** Raw values for UNC-29::GFP and UNC-63::GFP receptor cluster number.

DOI: https://doi.org/10.7554/eLife.35692.010

**Source data 2.** Raw values for ACR-12::GFP receptor cluster number and normalized UNC-29::GFP and UNC-63::GFP fluorescence intensity.

DOI: https://doi.org/10.7554/eLife.35692.011

**Figure supplement 1.** Mutations in specific AChR subunits and accessory proteins disrupt ACR-12 synaptic delivery and clustering.

DOI: https://doi.org/10.7554/eLife.35692.009

implicated in synapse formation, many of which have previously demonstrated expression in GABAergic neurons (*Cinar et al., 2005*) (*Figure 3—figure supplement 1*). Mutations in most genes tested (75%) produce no significant disruption in ACR-12 receptor clustering (*Figure 3A*, light blue). A second group, comprising 8 of the 36 genes analyzed (*Figure 3A*, green), produces mild to moderate (26–39%) decreases in ACR-12 clustering. Many of the mutants in this second group identify genes (e.g. *lev-10*, *madd-4*) that perform previously characterized functions in neuromuscular synapse development and muscle AChR clustering (*Gally et al., 2004*; *Pinan-Lucarré et al., 2014*) (*Figure 3—figure supplement 2A,B*), but appear to play comparatively minor roles in establishing synaptic connectivity with GABAergic neurons.

One of the 36 mutations tested is clearly distinguishable by a striking decrease in ACR-12 clustering. Mutation of the *nrx-1* gene (orange, *Figure 3A*) does not significantly affect transgene expression in GABAergic neurons (*Figure 3—figure supplement 2C,D*), but produces a ~70% reduction in ACR-12 receptor clustering (*nrx-1(ok1649)*, p<0.0001) (*Figure 3A,B*). *nrx-1* encodes the sole *C. elegans* ortholog of the synaptic organizer neurexin. Neurexin has been well documented to play roles in mammalian synapse formation and function (*Chen et al., 2017*; *Dean et al., 2003*; *Graf et al., 2004*; *Missler et al., 2003*). Roles for NRX-1 in *C. elegans* synapse formation remain, by comparison, less well defined. Moreover, roles for neurexin in establishing divergent connectivity have not been previously addressed in any system. Importantly, we do not observe appreciable alterations in the clustering of muscle AChRs in *nrx-1* mutants (*Figure 3C*), similar to previously reported findings (*Hu et al., 2012*). The profound alterations in ACR-12 localization described above, coupled with the lack of effect on muscle AChRs, therefore warranted an in-depth analysis of cholinergic synapses with GABAergic neurons in *nrx-1* mutants.

We first sought to distinguish whether *nrx-1* performs a specific role in synapse formation or serves more generalized functions in the developmental maturation of DD neurons. We studied the developmental remodeling of DD neurons, characterized by the dorsoventral repositioning of synaptic markers, which occurs at the L1/L2 transition in wild type animals (*Jin and Qi, 2018*; *White et al., 1978*). In particular, we examined the repositioning of pre- and post-synaptic markers expressed specifically in DD motor neurons (*Figure 3—figure supplement 3A*). Presynaptic remodeling, as measured using the synaptic vesicle marker mCherry::RAB-3, proceeds normally in *nrx-1* mutants, indicating that this process does not require neurexin expression. The clustering of ACR-12 receptors, however, is impaired both prior to and following remodeling in *nrx-1* mutants, suggesting that NRX-1 is required for receptor clustering in each of these developmental stages.

As is the case for mammals, the *C. elegans nrx-1* locus encodes both long (*nrx-1$_L$*) and short (*nrx-1$_S$*) neurexin isoforms (*Figure 3—figure supplement 3B*). The long isoform encodes a single pass transmembrane protein harboring intracellular PDZ binding and interleaved extracellular LNS (laminin-neurexin-sex hormone-binding globulin) and EGF-like domains (*Figure 3D*). The *ok1649* allele generates an in-frame deletion, eliminating 861 bp predicted to encode an extracellular LNS domain, raising the possibility that partial NRX-1 function may still be present in this strain. We therefore expanded our analysis to include additional *nrx-1* deletions, three of which (*ds1, tm1961,* and *nu485*) are predicted to primarily impact the long isoform, while another (*wy778*) removes the transmembrane and cytoplasmic domains shared by all NRX-1 isoforms (*Calahorro and Ruiz-Rubio, 2013*; *Maro et al., 2015*; *Tong et al., 2015*). All of the deletions tested disrupt ACR-12 receptor clustering in DD GABAergic neurons, with the most severe disruptions occurring in *nrx-1(wy778)* and

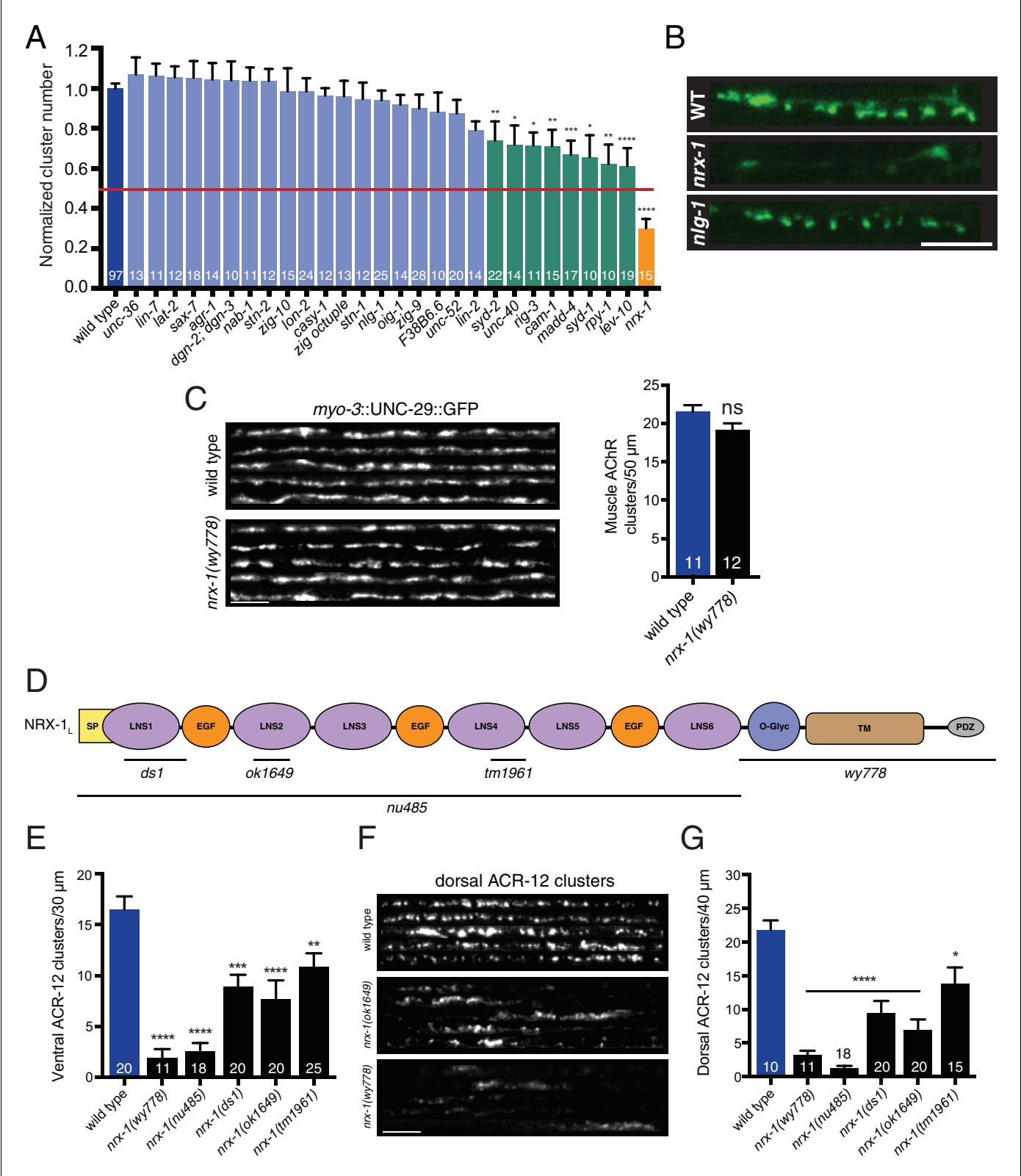

**Figure 3.** *nrx-1/n*eurexin is required for AChR localization in GABAergic motor neurons, but not muscles. (**A**) Quantification of ACR-12::GFP clusters in the synaptic region of the DD1 dendrite for the genotypes indicated, normalized to wild type. Red line indicates 50% reduction in puncta number. Colored bars indicate wild type (blue), no effect (light blue), modest effect (green), and severe clustering defects (orange). *p<0.05, **p<0.01, ***p<0.001, ****p<0.0001, ANOVA with Dunnett's multiple comparisons test. (**B**) Representative confocal images showing ACR-12::GFP clusters in the

*Figure 3 continued on next page*

Figure 3 continued

synaptic region of the DD1 dendrite for wild type (WT), *nrx-1(ok1649)*, and *nlg-1(ok259)* mutants. Scale bar, 5 μm. (C) Left, representative confocal images of the dorsal nerve cord region from five wild type and *nrx-1(wy778)* adult animals expressing *myo-3*::UNC-29::GFP to label muscle AChRs. Scale bar, 5 μm. Right, quantification of UNC-29::GFP clusters in a 50 μm region of the posterior dorsal nerve cord. (D) Domain structure of the NRX-1 long isoform (NRX-1$_L$). Deletion regions are indicated (black line). N-terminal signal peptide (SP), extracellular LNS, EGF, transmembrane (TM), PDZ binding (PDZ) domains, and O-linked glycosylation site are shown. (E) Quantification of ACR-12 receptor clusters (*unc-47*::ACR-12::GFP) in the ventral nerve cord for the genotypes indicated. **p<0.01, ***p<0.001, ****p<0.0001, ANOVA with Dunnett's multiple comparisons test. (F) Representative confocal images showing dorsal nerve cord ACR-12 receptor clusters for five wild type, *nrx-1(ok1649)*, and *nrx-1(wy778)* animals expressing *unc-47*::ACR-12::GFP. Scale bar, 5 μm. (G) Quantification of ACR-12 receptor clusters (*unc-47*::ACR-12::GFP) in the dorsal nerve cord for the genotypes indicated. *p<0.05, ****p<0.0001, ANOVA with Dunnett's multiple comparisons test.

DOI: https://doi.org/10.7554/eLife.35692.012

The following source data and figure supplements are available for figure 3:

**Source data 1.** Raw values for normalized ACR-12::GFP receptor clusters, muscle UNC-29::GFP receptor clusters, and ventral/dorsal ACR-12::GFP receptor clusters in *nrx-1* mutants.
DOI: https://doi.org/10.7554/eLife.35692.017

**Source data 2.** Raw values for UNC-29::GFP and UNC-63::GFP receptor cluster number, normalized wild type and *nrx-1* dendrite fluorescence, and SNB-1::GFP clusters.
DOI: https://doi.org/10.7554/eLife.35692.018

**Figure supplement 1.** Description of genes surveyed in candidate-based genetic screen.
DOI: https://doi.org/10.7554/eLife.35692.013

**Figure supplement 2.** LEV-10 and MADD-4 are required for muscle L-AChR clustering, and *nrx-1* deletion does not disrupt transgene expression.
DOI: https://doi.org/10.7554/eLife.35692.014

**Figure supplement 3.** Loss of functional *nrx-1* disrupts ACR-12 AChR localization, but *nrx-1* is not required for AChR membrane insertion or synaptic remodeling.
DOI: https://doi.org/10.7554/eLife.35692.015

**Figure supplement 4.** Neurexin is not essential for AMPAR localization or synaptic vesicle clustering in cholinergic motor neurons.
DOI: https://doi.org/10.7554/eLife.35692.016

*nrx-1(nu485)* mutants (*Figure 3E*). Strikingly, we also observe significant disruption of ACR-12 receptor clustering in the VD GABAergic neurons (*Figure 3F,G*) of *nrx-1* mutants, indicating a requirement for *nrx-1* at synapses onto both GABAergic neuron classes.

In some instances (14 of 34 animals for *nrx-1(wy778)*), we noted that a few ACR-12 receptor clusters remain detectable. We investigated whether these residual ACR-12 receptors are localized to the cell surface by injecting Alexa594 conjugated anti-HA antibody into *nrx-1* mutants expressing ACR-12::GFP::3xHA. Antibody fluorescent signal clearly colocalizes with ACR-12::GFP fluorescence, providing evidence that the few remaining receptor clusters in *nrx-1* mutants are present at the cell surface (*Figure 3—figure supplement 3C*). We interpret this result to indicate that neurexin is not essential for membrane insertion, although we can't rule out that this process may occur less efficiently in *nrx-1* mutants.

We next examined whether the localization of putative ACR-12 partnering subunits is also disrupted by mutation of *nrx-1*. As noted above, ACR-12 receptors are formed as heteromeric complexes in GABAergic neurons that likely incorporate the UNC-29 and UNC-63 AChR subunits. Neurexin deletion (*wy778*) reduces UNC-29::GFP and UNC-63::GFP clusters in DD1 by 60%, consistent with a requirement for neurexin in the proper localization of mature, heteromeric receptor complexes (*Figure 3—figure supplement 3D,E*). In contrast to our findings for GABAergic neurons, *nrx-1* deletion does not appreciably disrupt the clustering of AMPA-type glutamate receptors in interneurons (*Figure 3—figure supplement 4A*), consistent with the idea that neurexin is not globally required for the establishment of synaptic connectivity in worms. Additionally, we do not observe an appreciable decrease in cholinergic synaptic vesicle clusters (*acr-2*::SNB-1::GFP), although this analysis does not rule out all potential presynaptic defects (*Figure 3—figure supplement 4B*).

To elucidate mechanisms by which *nrx-1* may instruct the formation of synapses between cholinergic and GABAergic motor neurons, we evaluated mutations in the *nlg-1* gene. *nlg-1* encodes the sole *C. elegans* ortholog of neuroligin, a well characterized binding partner of neurexin (*Banerjee et al., 2017*; *Boucard et al., 2005*; *Hu et al., 2012*; *Ichtchenko et al., 1995*, *Ichtchenko et al., 1996*). We find that mutation of *nlg-1* produces no appreciable defects in ACR-12

receptor clustering (*Figure 3A,B*), indicating, surprisingly, that NRX-1 operates independently of NLG-1 to direct post-synaptic development in GABAergic neurons.

## Post-synaptic morphological development requires NRX-1

We next investigated whether neurexin is required for the outgrowth or stabilization of the spine-like processes we observe in DD dendrites. Wild type animals (at the L4 stage) have an average of 7–8 of these spiny protrusions within the synaptic region of the anterior DD1 dendrite (*Figure 4A,B*). In contrast, *nrx-1* mutants have strikingly reduced numbers of spiny protrusions. For example, only two spiny protrusions are visible on average in DD1 dendrites of *nrx-1(ok1649)*, and *nrx-1(wy778)* mutants show a near complete absence of spines (*Figure 4A,B*). *nrx-1* deletion disrupts both ACR-12 receptor clustering and spine outgrowth in posterior DD neurons to a similar extent as observed for DD1 (*Figure 4C*, *Figure 4—figure supplement 1A*), indicating that the requirement for *nrx-1* is shared across DD neurons. Roughly 15% of the receptor clusters remaining in *nrx-1(ok1649)* mutants are associated with the remaining spines (compared with 67% in wild type). In contrast, the number of dendritic receptor clusters is not appreciably altered by *nrx-1* deletion (p=0.67, student's t test), suggesting NRX-1 may preferentially regulate spine-associated receptor clusters in DD neurons. *nrx-1* deletion also significantly reduces spiny protrusions in L2 animals (the earliest stage at which they are visible) (*Figure 4—figure supplement 1B*), suggesting that neurexin is required for initial spine outgrowth, although an additional role in maintenance is also possible. In accordance with our finding that *nlg-1* is not required for ACR-12::GFP localization, mutation of *nlg-1* does not alter spine number (*Figure 4A,B*), arguing against an essential role for neuroligin in the formation of cholinergic synapses with GABAergic neurons. Likewise, spine number is not appreciably altered in either *acr-12* or *unc-63* mutants (*Figure 4A,B*), indicating that spine outgrowth proceeds normally in the absence of functional ACR-12 receptors.

## *nrx-1* is expressed and functionally required in cholinergic motor neurons

To understand how NRX-1 regulates the formation of cholinergic synapses with GABAergic neurons, we sought to define the requirements for *nrx-1* expression. We first examined expression of a *nrx-1$_L$*::GFP transcriptional reporter incorporating ~4.8 kb of sequence upstream of the *nrx-1$_L$* start site, and found that this reporter is strongly expressed in cholinergic motor neurons (*Figure 5A*). We next asked whether specific *nrx-1* expression in cholinergic neurons is sufficient to rescue the post-synaptic defects of *nrx-1* mutants. We found that cholinergic *nrx-1$_L$* expression reverses the ACR-12 clustering defects of *nrx-1(wy778)* mutants, while expression in either GABAergic neurons or muscles fails to rescue (*Figure 5B–D*). Notably, cholinergic expression of *nrx-1$_L$* also restores spine-like protrusions in *nrx-1* mutants (*Figure 5—figure supplement 1A,B*). Thus, our results suggest that pre-synaptic NRX-1 acts non-autonomously to direct post-synaptic assembly in GABAergic neurons. To investigate this possibility in more detail, we examined the subcellular localization of NRX-1 by expressing GFP-tagged NRX-1$_L$ (NRX-1$_L$::GFP) (*Maro et al., 2015*) in a subset of cholinergic neurons (DA/DB). Expression of *unc-129*::NRX-1$_L$::GFP produces discrete puncta along the dorsal nerve cord where the synaptic outputs of DA/DB neurons are located (*Figure 5E*). NRX-1$_L$::GFP clusters colocalize with clusters of mCherry::RAB-3 fluorescence, providing evidence that NRX-1$_L$ is preferentially localized to cholinergic presynaptic sites (*Figure 5E,F*). Intriguingly, cholinergic expression of a rescuing transgene lacking the PDZ binding motif located at the intracellular NRX-1 C-terminus (*nrx-1$_L$ΔPDZ*) in *nrx-1* mutants can also rescue ACR-12 clustering defects (*Figure 5—figure supplement 2A,B*). NRX-1$_L$ΔPDZ::GFP is exclusively expressed in cholinergic axons and partially colocalizes with mCherry::RAB-3 fluorescence, indicating that NRX-1 presynaptic localization and function is possible without PDZ protein binding via this motif (*Figure 5—figure supplement 2C–E*). Loss of *nlg-1* function does not affect the axonal localization of NRX-1$_L$::GFP (*Figure 5—figure supplement 3A,C–D*). NRX-1$_L$::GFP localization is also not appreciably altered by *acr-12* deletion, suggesting that the positioning of NRX-1 at presynaptic terminals occurs independently of post-synaptic receptor clustering (*Figure 5—figure supplement 3B–D*). Our results indicate NRX-1 positioning at presynaptic sites occurs independently of post-synaptic receptor localization, and raise the intriguing possibility that NRX-1 localization to the presynaptic domain may serve as an initiation signal for developmental maturation of post-synaptic specializations in GABAergic neurons.

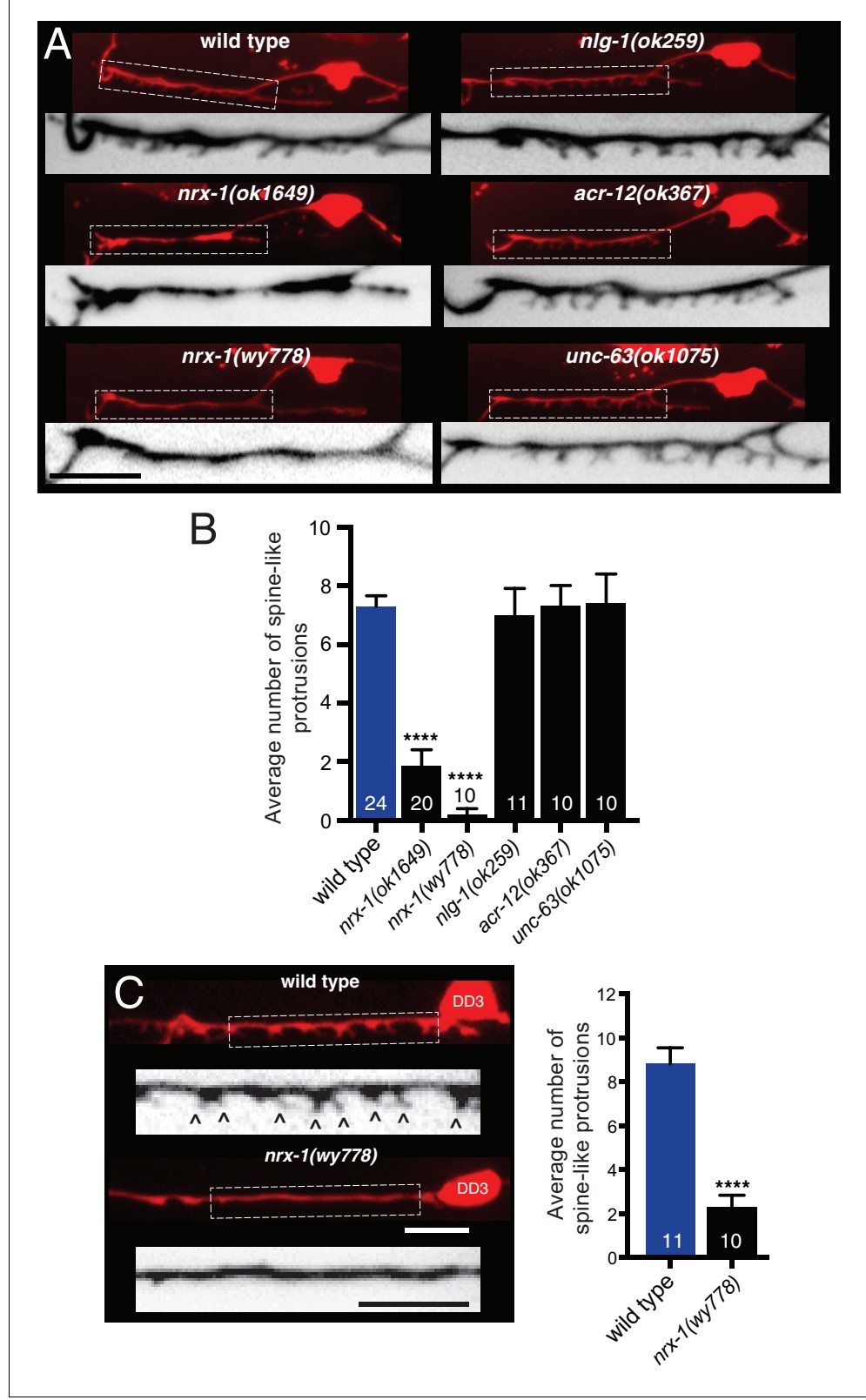

**Figure 4.** Synaptic architecture is dependent on *nrx-1*/neurexin. (A) Fluorescent confocal images of spine-like protrusions in the synaptic region of the DD1 dendrite (*flp-13*::mCherry) for the genotypes indicated. Inverted images show expanded views of the synaptic regions (indicated by dashed boxes in fluorescent images). Scale bar, 5 μm. (B) Quantification of spine-like protrusions in the DD1 dendrite for the genotypes indicated.

*Figure 4 continued on next page*

*Figure 4 continued*

****p<0.0001, ANOVA with Dunnett's multiple comparisons test. (**C**) Left, fluorescent confocal images of spine-like protrusions (arrowheads) in the DD3 neuron dendrite (*flp-13*::mCherry) for wild type and *nrx-1(wy778)* mutants. Inverted images show expanded views of the DD3 dendrites (indicated by dashed boxes in fluorescent images). Scale bars, 5 µm. Right, quantification of spine-like protrusions in the DD3 dendrite (25 µm region anterior to the cell body) for wild type and *nrx-1(wy778)* mutants. ****p<0.0001, student's t-test.
DOI: https://doi.org/10.7554/eLife.35692.019
The following source data and figure supplement are available for figure 4:

**Source data 1.** Raw values for spine-like protrusion number.
DOI: https://doi.org/10.7554/eLife.35692.021
**Source data 2.** Raw values for adult ACR-12::GFP receptor cluster number in DD3 and DD1 spine-like protrusion number at L2.
DOI: https://doi.org/10.7554/eLife.35692.022
**Figure supplement 1.** NRX-1 is required for ACR-12 receptor clustering and spine outgrowth in DD motor neurons.
DOI: https://doi.org/10.7554/eLife.35692.020

## The COE-type transcription factor *unc-3* directly controls neurexin expression

We have previously demonstrated that mutation of the COE-type (Collier/Olf/Ebf) transcription factor *unc-3* disrupts ACR-12 clustering in VD GABAergic neurons (*Barbagallo et al., 2017*). In light of our findings here that *nrx-1* deletion similarly disrupts ACR-12 clustering and spine-like protrusion outgrowth in DD neurons (*Figure 6A,B*), we next investigated the role of UNC-3 transcriptional regulation in the development of cholinergic connectivity with GABAergic DD neurons. Prior work has shown that activity of UNC-3 is essential for the specification of cholinergic neurotransmitter identity (*Kratsios et al., 2015*, *2011*). To investigate the requirement for *unc-3* in ACR-12 clustering, we first evaluated whether cholinergic transmission itself is critical for the development of post-synaptic specializations on DD neurons. We found that mutations in the *unc-17* cholinergic vesicular ACh transporter produces no appreciable changes in spine-like protrusion number or ACR-12 clusters (*Figure 6B*), arguing against a strong requirement for cholinergic transmission in the formation of these structures.

We next asked whether UNC-3 transcriptional regulation of *nrx-1* expression is critical for the development of cholinergic connectivity with GABAergic neurons. To address this question, we first tested whether *unc-3* is required for expression of the *nrx-1$_L$*::GFP transcriptional reporter described above. We found that mutation of *unc-3* significantly reduces *nrx-1$_L$*::GFP fluorescence in motor neuron cell bodies of the ventral nerve cord, as well as the majority of fluorescence in ventral cord processes (*Figure 6C,D*). The remaining ventral cord GFP fluorescence is associated with the processes of head neurons that project into the nerve cord, which are presumably not subject to *unc-3* regulation. We next used fluorescent in situ hybridization (FISH) to determine the effects of *unc-3* mutation on *nrx-1* mRNA abundance. Fluorescent signals indicating *nrx-1* mRNA are clearly associated with cholinergic motor neuron cell bodies (co-labeled with *unc-17*::GFP) in wild type animals (*Figure 6E*), consistent with our prior analysis using the *nrx-1$_L$*::GFP transcriptional reporter. The smFISH signals are strongly diminished in *nrx-1(nu485)* deletion mutants (*Figure 6E,F*), confirming they accurately report *nrx-1* mRNA abundance. Labeled *nrx-1* mRNA signals in cholinergic motor neurons are strikingly reduced by mutation of *unc-3*, consistent with the possibility that *nrx-1* is a transcriptional target of *unc-3* (*Figure 6E,F*).

We noted that a second *nrx-1$_L$* transcriptional reporter incorporating only ~2 kb of *nrx-1* regulatory sequence did not produce strong fluorescence in ventral cord motor neurons (*Figure 6C,D*). We reasoned that regulatory elements required for *nrx-1* expression in these neurons may be present in the sequence that differs across these two transcriptional reporters (2 versus 4.8 kb) (*Figure 6C,D*). As both mammalian COE transcription factors and UNC-3 bind a conserved COE binding motif (TCCCNN$^G/_A$$^G/_A$$^G/_A$) to regulate transcription of target genes (*Kim et al., 2005*; *Kratsios et al., 2011*; *Wang et al., 2015*, *1993*), we searched for COE binding motifs within this region. We identified a potential COE motif (TCCCAAAGGG) located approximately 20 bp from the 5′ end of the 4.8 kb *nrx-1$_L$*::GFP transcriptional reporter. Mutation of this site

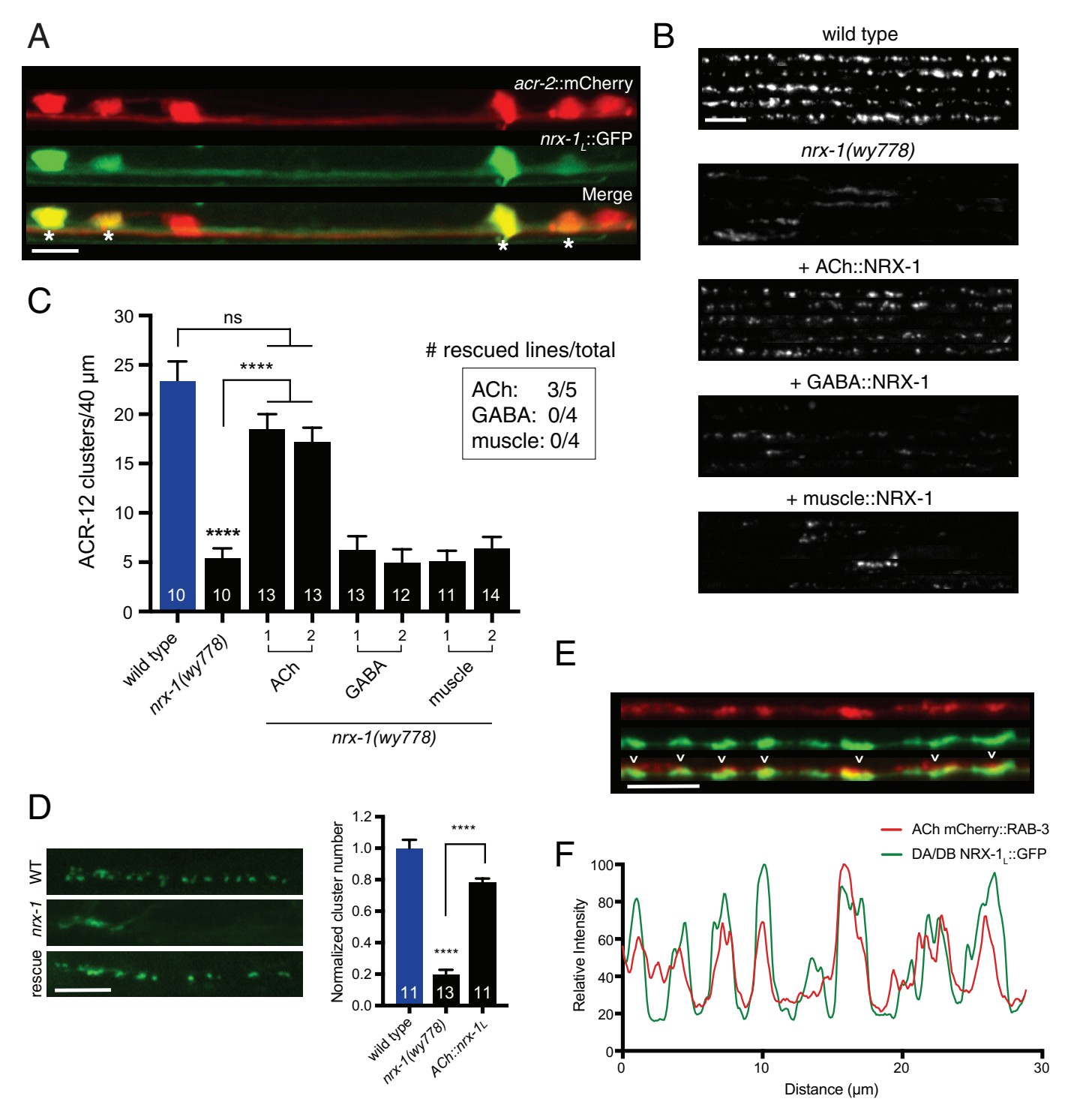

**Figure 5.** Cell-specific expression of neurexin in cholinergic motor neurons restores ACR-12 localization to synapses. (**A**) Confocal images of the ventral nerve cord in a transgenic strain expressing *nrx-1*$_L$::GFP together with the cholinergic motor neuron marker *acr-2*::mCherry. Asterisks indicate coexpression. Scale bar, 5 μm. (**B**) Confocal images of the dorsal nerve cord (*unc-47*::ACR-12::GFP) for the genotypes indicated. For each, five representative images are shown. In B and C, rescue was evaluated by cholinergic (*unc-17β* promoter), GABAergic (*unc-47* promoter), or muscle-specific (*myo-3* promoter) expression of a NRX-1$_L$ minigene in *nrx-1(wy778)* mutants. Scale bar, 5 μm. (**C**) Quantification of ACR-12::GFP receptor clusters in the dorsal nerve cord for the genotypes indicated. ****p<0.0001, ns = not significant, ANOVA with Tukey's multiple comparisons test. Two transgenic lines are shown for each rescue construct. Inset, number of rescuing lines/total transgenic lines tested for each construct. (**D**) Left, confocal

*Figure 5 continued on next page*

*Figure 5 continued*

images of ACR-12::GFP clusters in the DD1 synaptic region for the genotypes indicated. Rescue refers to cholinergic expression (*ufEx1114*, line #1 in *Figure 5C*) of NRX-1$_L$ in *nrx-1(wy778)* mutants. Scale bar, 5 µm. Right, quantification of ACR-12::GFP clusters in the DD1 synaptic region for the genotypes indicated, normalized to control (*ufEx441*). ****p<0.0001, ANOVA with Tukey's multiple comparisons test. (E) Confocal images of the dorsal nerve cord in an adult animal expressing NRX-1$_L$::GFP (*unc-129* promoter) with mCherry::RAB-3 (*acr-2* promoter). Colocalization is indicated by arrowheads. Scale bar, 5 µm. (F) Line scans showing relative fluorescent intensity of NRX-1$_L$::GFP (green) and mCherry::RAB-3 (red) for a 30 µm region of the dorsal nerve cord.

DOI: https://doi.org/10.7554/eLife.35692.023

The following source data and figure supplements are available for figure 5:

**Source data 1.** Raw values for ACR-12::GFP receptor cluster number with expression of neurexin rescuing constructs and for colocalization of mCherry::Rab-3 and NRX-1::GFP fluorescence.

DOI: https://doi.org/10.7554/eLife.35692.027

**Source data 2.** Raw values for spine-like protrusions and ACR-12::GFP receptor clustering with rescue constructs, and for NRX-1::GFP fluorescence in indicated mutants.

DOI: https://doi.org/10.7554/eLife.35692.028

**Figure supplement 1.** NRX-1 acts presynaptically to regulate spine outgrowth.

DOI: https://doi.org/10.7554/eLife.35692.024

**Figure supplement 2.** NRX-1 protein lacking the PDZ binding motif rescues ACR-12 receptor clustering defects in *nrx-1* mutants and localizes to cholinergic axon terminals.

DOI: https://doi.org/10.7554/eLife.35692.025

**Figure supplement 3.** Loss of *nlg-1* or *acr-12* does not affect NRX-1 localization to cholinergic axons.

DOI: https://doi.org/10.7554/eLife.35692.026

(T<u>CCC</u>AAAGGG >>T<u>AAAAAA</u>GGG) within the 4.8 kb *nrx-1$_L$*::GFP transcriptional reporter eliminates all fluorescence from ventral cord motor neurons, while fluorescence in the processes extending from head neurons remains visible (*Figure 6C,D*), offering evidence that UNC-3 directly regulates *nrx-1* transcription in ventral cord neurons.

We reasoned that forced expression of *nrx-1$_L$* in cholinergic neurons using a promoter not subject to *unc-3* regulation may allow NRX-1 to coordinate synapse development independently of UNC-3 transcriptional regulation. We expressed the *nrx-1$_L$* isoform using a regulatory region of the *unc-3* gene that drives expression in ventral cord cholinergic neurons (*Barbagallo et al., 2017*). We found that cholinergic-specific expression of the *nrx-1$_L$* isoform significantly restored receptor clusters in the dendritic region of *unc-3* animals (*Figure 6G*), indicating that the lack of ACR-12 receptor clusters in *unc-3* mutants is largely driven by the absence of *nrx-1* expression, although additional phenotypes associated with mutation of *unc-3* may contribute (e.g. variable nerve cord defasciculation) (*Barbagallo et al., 2017*). These findings define the gene regulatory mechanisms controlling neurexin expression in presynaptic neurons and illustrate their involvement in the establishment of synaptic connectivity.

## *nrx-1* deletion impairs cholinergic neurotransmission onto GABAergic neurons

To investigate how *nrx-1* deletion impacts the spatial arrangement of pre- and post-synaptic specializations, we examined strains coexpressing ACR-12::GFP in GABAergic neurons with the synaptic vesicle marker mCherry::RAB-3 in cholinergic neurons. In the wild type, ACR-12 receptor clusters at the tips of spiny protrusions are submerged within the presynaptic domains of cholinergic axons, where synaptic contacts are presumably located (*Figure 7A*, left). In *nrx-1* mutants, however, we noted a gap between the neurites of the pre- and post-synaptic neurons (*Figure 7A*, right), suggesting that *nrx-1* coordinates the extension of receptor-bearing spiny protrusions to presynaptic domains of cholinergic axons. These results, in combination with the receptor clustering defects in VD GABAergic neurons and the lack of an appreciable effect on muscle synapses described above, predict that *nrx-1* deletion would impair cholinergic synaptic activation of GABAergic neurons, while cholinergic transmission onto muscles would remain unaffected.

We recorded Ca$^{2+}$ transients from either GABAergic motor neurons or muscles immediately following presynaptic cholinergic depolarization in order to address this question (*Figure 7—figure supplement 1A*). We used combined cell-specific expression of Chrimson for cholinergic

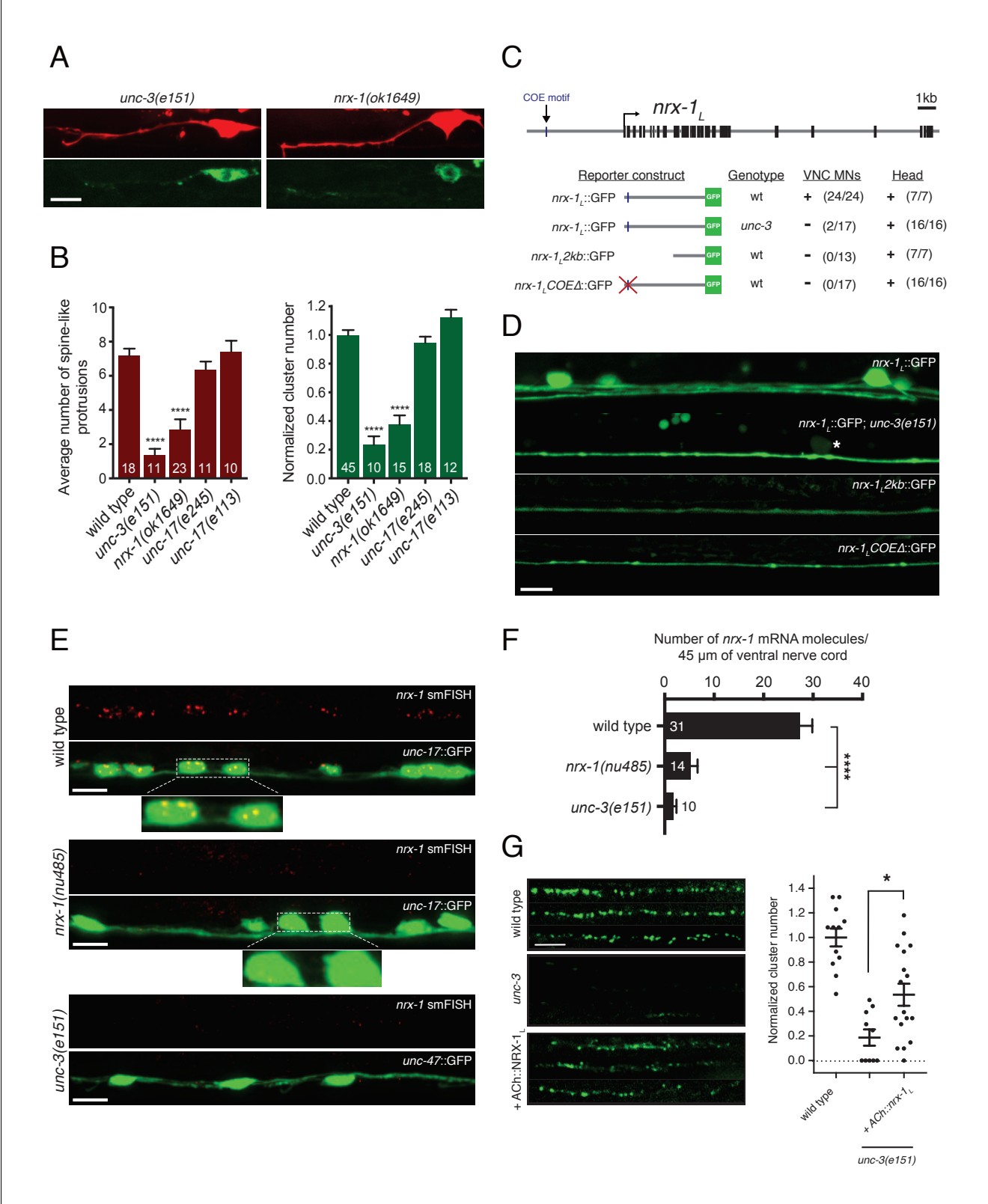

**Figure 6.** Transcriptional control of neurexin expression in ACh motor neurons by UNC-3. (**A**) Confocal images of the DD1 synaptic region and cell soma in *nrx-1(ok1649)* and *unc-3(e151)* mutants expressing *flp-13*::mCherry (upper) or *flp-13*::ACR-12::GFP (lower). Note the absence of spine-like protrusions and ACR-12 receptor clusters for both genotypes. Although we also note variable defects in neurite extension in *unc-3* mutants, reductions in spine-like protrusions and ACR-12 clusters are clearly evident when neurite extension appears unaffected. Notably, *flp-13*::mCherry fluorescence is

*Figure 6 continued on next page*

*Figure 6 continued*

not altered by mutation of *unc-3*, arguing against *unc-3* regulation of the *flp-13* promoter. (B) Left, quantification of spine-like protrusions in the DD1 dendrite for the genotypes indicated. ****p<0.0001, ANOVA with Dunnett's multiple comparisons test. Right, quantification of ACR-12::GFP clusters in the DD1 synaptic region for the genotypes indicated, normalized to control (*ufIs126*). ****p<0.0001, ANOVA with Dunnett's multiple comparisons test. (C) Mutational analysis of the regulatory region of the *nrx-1$_L$* promoter. Upper, genomic organization of the *nrx-1* locus. Black boxes, exons. Blue line, COE motif upstream of the *nrx-1* start site. Lower, schematics of promoter regions fused with GFP (green) corresponding to the images in D. (+) indicates strong expression in ventral nerve cord or head neurons of fourth larval stage animals, (-) indicates lack of expression. Number of animals with GFP expression in either ventral nerve cord or head neurons is indicated in parentheses. (D) Confocal images of *nrx-1$_L$*::GFP expression in the ventral nerve cord for the genotypes indicated. Note the decreased *nrx-1$_L$*::GFP fluorescence in *unc-3(e151)* mutants, with expression from a truncated (2 kb) promoter (*nrx-1$_L$2kb*), or with disruption of the COE motif (*nrx-1$_L$COEΔ*). The remaining fluorescent signal is associated with processes originating from head neurons. (E) Fluorescent in situ hybridization (FISH) signals indicating *nrx-1* mRNA abundance in wild type, *nrx-1(nu485)*, and *unc-3(e151)* mutant animals expressing *unc-17*::GFP or *unc-47*::GFP to visualize the ventral nerve cord. mRNA molecules are labeled by CAL Fluor Red 590 Dye conjugated probes and appear as red dots. Insets, expanded views of mRNA labeling associated with cholinergic motor neuron cell bodies. (F) Quantification of *nrx-1* mRNA molecules per 45 μm segment of the ventral nerve cord for the genotypes indicated. ****p<0.0001, ANOVA with Dunnett's multiple comparisons test. (G) Left, representative confocal images showing ACR-12::GFP clusters in the dorsal nerve cord for three wild type, *unc-3(e151)* or *unc-3* rescue animals expressing *unc-47*::ACR-12::GFP. Rescue refers to cholinergic-specific (*unc-3* promoter) expression of NRX-1$_L$ in *unc-3(e151)* mutants (*ACh::nrx-1$_L$*). Right, scatterplot of ACR-12 receptor cluster number in a 40 μm region of the dorsal nerve cord for the genotypes indicated, normalized to control. n ≥ 10 for each genotype. *p<0.05, ANOVA with Tukey's multiple comparisons test. Scale bars, 5 μm (A, D–E, G).
DOI: https://doi.org/10.7554/eLife.35692.029

The following source data is available for figure 6:

**Source data 1.** Raw values for spine-like protrusion and ACR-12::GFP receptor cluster number, and for *nrx-1* mRNA FISH signal.
DOI: https://doi.org/10.7554/eLife.35692.030

depolarization (*Klapoetke et al., 2014*; *Larsch et al., 2015*), and GCaMP6s for monitoring [Ca$^{2+}$] changes (*Chen et al., 2013*) in either post-synaptic GABAergic motor neurons or body wall muscles (*Figure 7B,E*). Strikingly, we found that *nrx-1* deletion disrupts GABA neuron Ca$^{2+}$ transients in response to cholinergic stimulation, but produces no appreciable effect on muscle Ca$^{2+}$ transients, consistent with a specific requirement for *nrx-1* in the development of functional connectivity between cholinergic and GABAergic neuron, but not muscle, synaptic partners (*Figure 7B–G*).

Cholinergic depolarization (5 s) evokes robust stimulus-coupled Ca$^{2+}$ transients in both GABAergic neurons (67% of stimuli) and muscles (88% of stimuli) that occur within 250 ms of stimulus onset (average response latency: $0.22 \pm 0.06$ s in motor neurons and $0.25 \pm 0.02$ s in muscles). These transients are not observed in the absence of cholinergic Chrimson expression or in the absence of retinal (*Figure 7—figure supplement 1B,C*), consistent with a requirement for presynaptic Chrimson-mediated depolarization. In both cell types, evoked Ca$^{2+}$ transients rise rapidly following stimulation ($\tau_{rise}$: $0.48 \pm 0.08$ s in GABA neurons; $0.74 \pm 0.06$ s in muscles), and persist throughout the duration of stimuli before decaying to baseline. Motor neuron transients are typically shorter in duration (mean duration: $7.7 \pm 1$ s in GABA neurons; $10.8 \pm 0.5$ s in muscles) and decay more rapidly ($\tau_{decay}$: $1.1 \pm 0.4$ s in GABA neurons; $4.6 \pm 0.6$ s in muscles) compared with muscle transients, likely reflecting differences in both synaptic connectivity and physiology across the two cell types.

For both GABA neurons and muscles, evoked Ca$^{2+}$ responses are eliminated almost completely by mutations that impair post-synaptic AChR function in the respective cell types (*acr-12* or *unc-29*; *acr-16*, respectively). Specifically, *acr-12* deletion reduces the mean peak amplitude of GABA neuron calcium responses to cholinergic stimulation by 60% (*Figure 7C–D*, *Figure 7—figure supplement 1D*) and increases the failure rate (no response to stimulation) by 53% compared to wild type, consistent with prior electrophysiology studies (*Petrash et al., 2013*). Similarly, for muscles, the mean peak amplitude of calcium responses to cholinergic stimulation is reduced by 95% in *unc-29;acr-16* double mutants (*Figure 7F–G*, *Figure 7—figure supplement 1E*), and the failure rate is increased roughly 15-fold to 77%, consistent with prior electrophysiology studies of evoked synaptic responses in these double mutants (*Francis et al., 2005*). *nrx-1* deletion reduces the mean peak amplitude of GABA neuron calcium responses to cholinergic stimulation by roughly 71% (*Figure 7C–D*, *Figure 7—figure supplement 1D*), and increases the failure rate for GABA neuron recordings by 47%. By comparison, *nrx-1* deletion does not produce a significant decrease in either mean peak fluorescence (*Figure 7F–G*, *Figure 7—figure supplement 1E*) or the failure rate in recordings of evoked muscle activity. Together, these findings support a specific requirement for *nrx-1* in cholinergic transmission

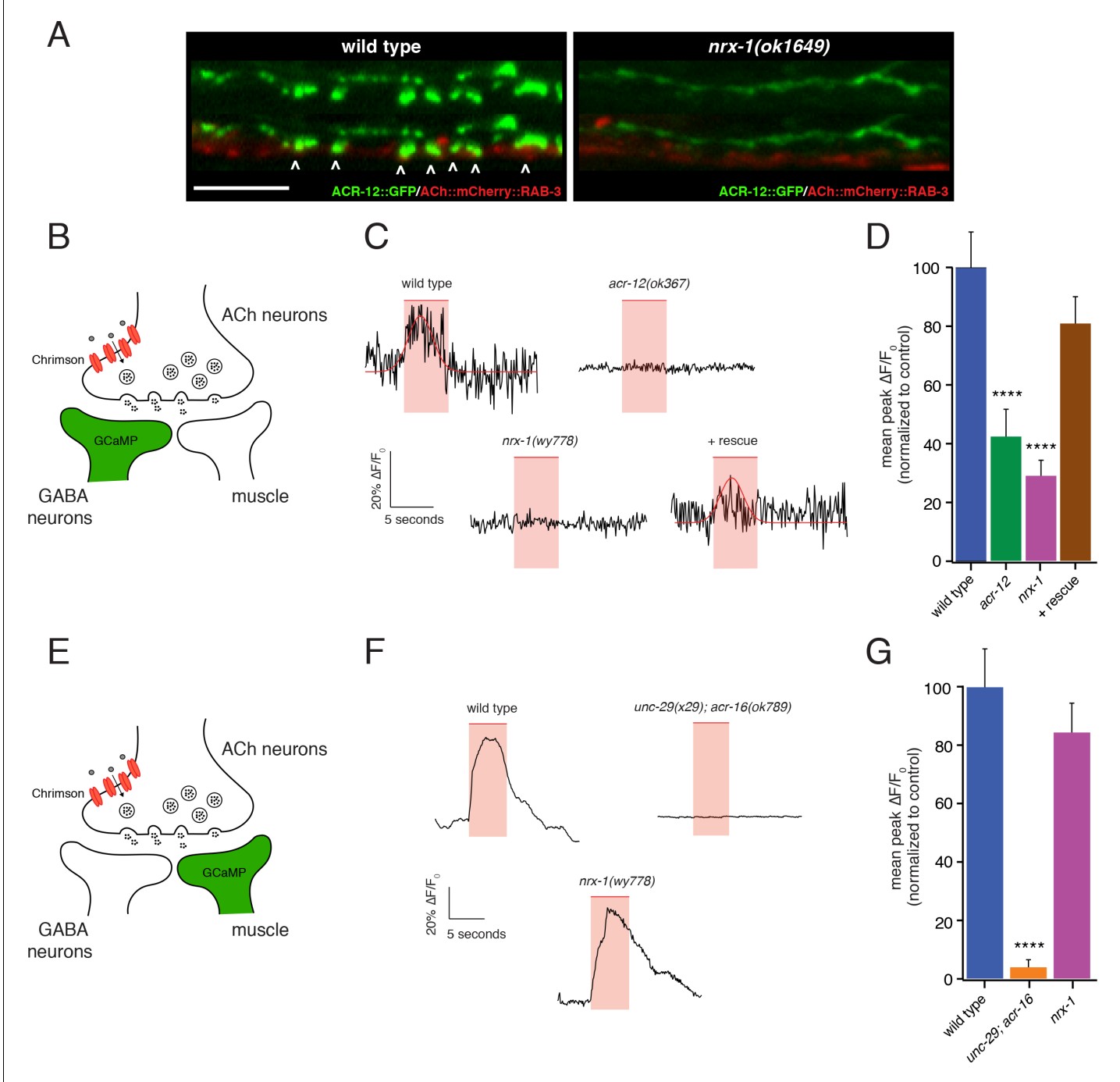

**Figure 7.** *nrx-1* mutants show functional defects in synaptic connectivity. (**A**) Confocal images of pre-synaptic (*acr-2*::mCherry::RAB-3) and post-synaptic (*flp-13*::ACR-12::GFP) specializations in the DD1 synaptic region for the genotypes indicated. Arrowheads indicate receptor-bearing spiny protrusions extending into the presynaptic region of cholinergic axons in wild type. Note gap between the pre- and post- synaptic regions in *nrx-1(ok1649)* mutants. Scale bar, 5 μm. (**B**) Cartoon depicting specific expression of Chrimson (*acr-2*::Chrimson) in cholinergic neurons together with GCaMP (*ttr-39*::GCaMP6s::SL2::mCherry, green) in GABAergic neurons, applies to C-D. (**C**) Representative calcium transients in GABAergic motor neurons evoked by light stimulation (red shaded region) of cholinergic neurons for the genotypes indicated. Rescue refers to cholinergic expression of NRX-1$_L$ in *nrx-1 (wy778)* mutants (**C–D**). Red line indicates Gaussian fit to the wild type and rescue traces. (**D**) Quantification of the mean peak $\Delta F/F_0$ upon Chrimson stimulation for the genotypes indicated, normalized to control (*ufIs155; ufIs157*). ****$p<0.0001$, ANOVA with Dunnett's multiple comparisons test. (**E**) Cartoon depicting specific expression of Chrimson (*acr-2*::Chrimson) in cholinergic neurons together with GCaMP (*myo-3*::NLSwCherry::SL2::GCaMP6s, green) in muscles, applies to F-G. (**F**) Representative calcium transients in muscle cells evoked by light stimulation (red shaded region) of cholinergic

*Figure 7 continued on next page*

*Figure 7 continued*

motor neurons for the genotypes indicated. (**G**) Quantification of the mean peak $\Delta F/F_0$ upon Chrimson stimulation for the genotypes indicated, normalized to control (*zfEx813; ufIs157*). ****$p<0.0001$, ANOVA with Dunnett's multiple comparisons test.

DOI: https://doi.org/10.7554/eLife.35692.031

The following source data and figure supplement are available for figure 7:

**Source data 1.** Raw values for calcium imaging peak fluorescence and normalized peak fluorescence.

DOI: https://doi.org/10.7554/eLife.35692.033

**Source data 2.** Raw values for body bend amplitude ratio during movement.

DOI: https://doi.org/10.7554/eLife.35692.034

**Figure supplement 1.** *nrx-1* mutants have defects in transmission onto GABAergic neurons and abnormalities in dorsoventral bending.

DOI: https://doi.org/10.7554/eLife.35692.032

onto GABA neurons, while *nrx-1* appears dispensable for transmission onto muscles under our recording conditions. Notably, both the failure rate and mean peak amplitude of evoked GABA neuron $Ca^{2+}$ responses are restored to wild type levels with expression of a rescuing *nrx-1_L* transgene in *nrx-1* mutants using a cholinergic neuron-specific promoter (*Figure 7C–D*, *Figure 7—figure supplement 1D*).

Consistent with a requirement for NRX-1 in cholinergic synaptic connectivity with GABAergic motor neurons, automated worm track analysis showed that *nrx-1* mutants display defects in the amplitude of dorsoventral bending, a feature of worm movement previously associated with GABAergic function (*McIntire et al., 1993*; *Petrash et al., 2013*). These effects are rescued with cell-specific expression of *nrx-1_L* in cholinergic neurons (*Figure 7—figure supplement 1F*). Thus, presynaptic *nrx-1* expression in cholinergic neurons is required for synaptic connectivity between cholinergic and GABAergic motor neurons, and deficits in these connections alter motor performance. Together, our data indicate that NRX-1 located in presynaptic cholinergic neurons is required for establishing synaptic connectivity with partnering GABAergic neurons, but not muscle cells. NRX-1 signaling promotes both receptor clustering and the outgrowth of post-synaptic spine-like morphological features in GABAergic dendrites. Our findings support a model where distinct synaptic organizers, acting on specific post-synaptic targets, are coordinately regulated with neuronal identity, perhaps offering a mechanism for independent developmental regulation of synaptic outputs across alternate partners (*Figure 8*).

## Discussion

Neurons often make divergent synaptic connections onto multiple postsynaptic partners, and these connections are critical for proper neural circuit performance in the brain. In this study, we use dyadic *C. elegans* synapses between cholinergic motor neurons and their muscle and GABAergic motor neuron postsynaptic partners as a model to define novel molecular mechanisms controlling divergent connectivity. First, we identify spine-like dendritic specializations on GABAergic DD neurons. We find that heteromeric post-synaptic AChR complexes composed of the ACR-12, UNC-63, UNC-38, UNC-29 and LEV-1 subunits are localized to these structures. Second, we identify a novel neurexin signaling pathway required both for the formation of post-synaptic specializations and for AChR clustering. In contrast, *nrx-1*/neurexin is not required for the development of cholinergic synapses onto muscles, which instead require molecularly distinct pathways that have been described previously (*Francis et al., 2005*; *Gally et al., 2004*; *Gendrel et al., 2009*; *Jensen et al., 2012*; *Pinan-Lucarré et al., 2014*; *Rapti et al., 2011*). Third, we find that presynaptic *nrx-1* expression in cholinergic neurons is required for the establishment of synapses with GABAergic neurons, and transcriptional regulation of *nrx-1* is achieved through actions of the COE transcription factor *unc-3*. These findings suggest a model where cholinergic expression of *nrx-1* initiates synapse formation with GABAergic neurons via NRX-1 mediated trans-synaptic signaling. Finally, the ACh-GABA connectivity defects that we observe in *nrx-1* mutants are paralleled by significant impairment of evoked cholinergic transmission onto GABAergic neurons, and altered sinusoidal movement. In contrast, functional connectivity with muscles is unaffected. Together, our findings provide evidence that distinct molecular signaling pathways act in parallel to establish divergent connections at dyadic synapses in the motor circuit, raising the interesting possibility that differential use of synaptic organizers

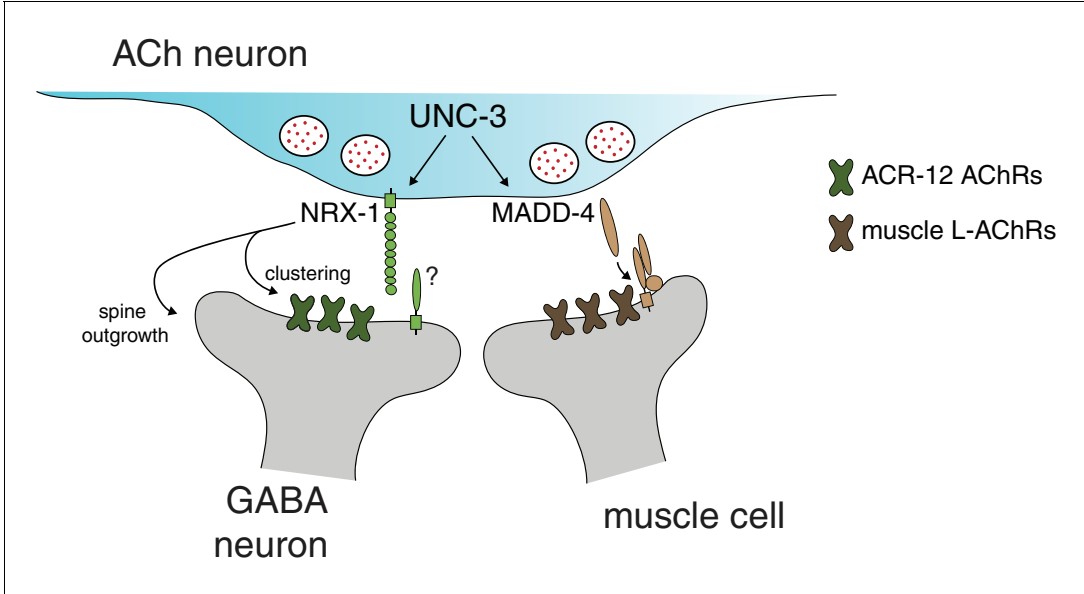

**Figure 8.** Distinct molecular scaffolds direct partner-specific connectivity. Distinct molecular scaffolds coordinate post-synaptic development in GABAergic neurons vs muscle. NRX-1/neurexin located at sites of presynaptic cholinergic release acts to coordinate ACR-12 receptor localization (green) and spine outgrowth in GABA neurons. A complex of proteins direct receptor clustering (brown) at the neuromuscular junction (*Gally et al., 2004*; *Gendrel et al., 2009*; *Pinan-Lucarré et al., 2014*; *Rapti et al., 2011*). NRX-1 and MADD-4 (*Kratsios et al., 2015*) expression in cholinergic neurons is transcriptionally co-regulated with neurotransmitter identity by the COE transcription factor UNC-3.

DOI: https://doi.org/10.7554/eLife.35692.035

may be similarly utilized in the brain to provide a molecular code directing divergent connectivity (*Figure 8*).

## Heteromeric ACR-12 receptor complexes cluster in specialized post-synaptic domains of GABAergic DD dendrites

Using candidate deletion analysis and cell-specific rescue, we identified AChR subunits and accessory proteins required for the assembly and localization of ACR-12 receptors in GABAergic neurons, defining the subunit composition of this neuronal receptor. Our findings implicate four additional receptor subunits (UNC-38, UNC-63, UNC-29, LEV-1) that co-assemble with ACR-12 to form pentameric receptor complexes in GABAergic neurons, and demonstrate that three accessory proteins (UNC-74, UNC-50, RIC-3) with more generalized roles in AChR assembly and maturation (*Boulin et al., 2008*; *Halevi et al., 2002*; *Jospin et al., 2009*) are also required. These findings provide evidence that ACR-12 receptors in GABA neurons are similar in subunit composition to muscle L-AChRs, differing only in the inclusion of the ACR-12 subunit in GABA neurons, whereas muscle L-AChRs incorporate the LEV-8 subunit (*Boulin et al., 2008*; *Towers et al., 2005*). Consistent with our analysis, a prior study showed that ACR-12 can be co-purified with the UNC-29 or LEV-1 subunits (*Gottschalk et al., 2005*). We have previously shown that *unc-29* is expressed in GABAergic motor neurons, and UNC-29::GFP localizes similarly to ACR-12::GFP in DD neurons, both in the mature animal and during developmental remodeling of these neurons (*He et al., 2015*).

We show that ACR-12 receptor complexes are concentrated at the tips of spine-like dendritic protrusions in GABAergic DD neurons. Spiny processes associated with D-type GABAergic neurons had been noted in prior electron microscopy studies (*White et al., 1976*; *1986*), but, to our knowledge, were not characterized further. We find that these AChR-containing dendritic protrusions are apposed by presynaptic clusters of cholinergic vesicles and increase in number during the course of larval development, perhaps representing new synaptic connections formed with post-embryonic born cholinergic neurons that are integrated into the circuit following the L1/L2 transition (*White et al., 1978*). While further investigation of these structures will undoubtedly reveal additional insights, the characteristics we define here raise the interesting possibility that these dendritic

protrusions are structural specializations for housing neurotransmitter receptors and other proteins required for post-synaptic signaling, perhaps representing an evolutionary precursor to mammalian dendritic spines.

## The synaptic organizer neurexin directs post-synaptic development in a partner-specific manner

*nrx-1* deletion impairs both AChR localization and spiny outgrowths in DD neurons, creating a gap between the pre- and post-synaptic neurons. Similarly, in *Drosophila,* mutation of the single neurexin gene *dnrx* causes disorganization of synaptic structure at the neuromuscular junction (*Li et al., 2007*). Knockout of two out of the three mouse alpha neurexins reduces dendrite length and total spine number in the cortex (*Dudanova et al., 2007*), though significant numbers of dendritic spines remain detectable. In our studies, we find that *nrx-1* is required for ACR-12 receptor localization in GABAergic DD and VD neurons, although we observe spine-like protrusions only in DD neurons. These findings may point toward the idea that NRX-1 is not solely involved in directing spine development, but serves an additional role in receptor clustering. Further, we show that dendritic protrusions form independently of a requirement for AChRs containing either ACR-12 or UNC-63, offering additional support that spine outgrowth and receptor clustering may be independently regulated by NRX-1. Similarly, dendritic spines on mouse CA1 pyramidal neurons form normally in the absence of functional glutamate receptors (*Lu et al., 2013*).

Although GABA neuron ACR-12 AChRs and muscle L-AChRs share very similar subunit composition (*Boulin et al., 2008*; *Lewis et al., 1980*), we find that genes required for proper localization of muscle L-AChRs (e.g. *madd-4, lev-10*) play comparatively minor roles at synapses onto GABA neurons. Conversely, loss of *nrx-1* function specifically affects GABAergic, but not muscle, AChR clustering, and cholinergic transmission onto muscles appears largely unaffected. Our cell-specific rescue experiments indicate *nrx-1* acts in cholinergic neurons to coordinate post-synaptic development in GABAergic neurons via trans-synaptic signaling.

Numerous studies support neuroligin as a primary trans-synaptic binding partner with neurexin (*Boucard et al., 2005*; *Comoletti et al., 2006*; *Ichtchenko et al., 1995*, *Ichtchenko et al., 1996*; *Nguyen and Südhof, 1997*). Indeed, *C. elegans* NRX-1/neurexin function has been characterized almost exclusively in the context of its partnership with NLG-1/neuroligin. A retrograde neurexin-neuroligin signaling pathway that regulates neurotransmitter release has been described, involving signaling through the $Ca^{2+}$ channel auxiliary subunit UNC-36/$\alpha 2\delta$ (*Hu et al., 2012*; *Tong et al., 2017*). Consistent with this representing a distinct mechanism from that described in our work, mutation of *unc-36* has no appreciable effect on post-synaptic development in our experiments. Postsynaptic expression of NLG-1/neuroligin is required at GABAergic synapses. However, MADD-4/Punctin likely acts as a major presynaptic partner for NLG-1 in this case, with NRX-1 playing a comparatively minor role (*Maro et al., 2015*; *Tong et al., 2015*; *Tu et al., 2015*). Finally, recent work also implicates neurexin-neuroligin signaling in sexually dimorphic neurite plasticity (*Hart and Hobert, 2018*). In contrast to these studies, we find that NRX-1 operates independently of NLG-1 to direct the formation of cholinergic synapses with GABAergic neurons. How therefore might presynaptic neurexin direct postsynaptic maturation? Our analysis shows that presynaptic NRX-1 localizes properly in the absence of *acr-12,* arguing against a requirement for direct binding of neurexin to the postsynaptic receptor. Prior studies offer strong evidence for alternate neurexin binding partners that support trans-synaptic signaling (*Boucard et al., 2012*; *de Wit et al., 2009*; *Ko et al., 2009*; *Missler et al., 1998*; *Petrenko et al., 1996*; *Pettem et al., 2013*; *Sugita et al., 2001*; *Uemura et al., 2010*). For several of these gene families (e.g. neurexophilins, cerebellins), clear *C. elegans* orthologs are not present. Others are included in our candidate analysis (e.g. *casy-1*/calsyntenin, *lat-2*/latrophilin), but single gene mutations do not produce post-synaptic defects comparable to mutation of *nrx-1*, suggesting either the possibility of a novel post-synaptic *nrx-1* binding partner or redundant post-synaptic mechanisms. Additional genetic or proteomic studies will be required to distinguish between these possibilities and address this important question.

## Transcriptional control of partner-specific synaptic connectivity

Transcriptional regulators of neurexin expression are only beginning to be elucidated (*Runkel et al., 2013*). Previously, we found that mutation of the COE-type transcription factor *unc-3* disrupts AChR

clustering in GABAergic dendrites (*Barbagallo et al., 2017*). These disrupted clusters are unlikely to reflect a requirement for acetylcholine release in AChR clustering, as neither tetanus toxin expression nor mutation of *unc-17*/vAChT produces appreciable defects in ACR-12 clustering (*Barbagallo et al., 2017*) (this study). Here, we find that neurexin is a transcriptional target of UNC-3, and we propose that UNC-3 regulation of *nrx-1* expression directs development of postsynaptic specializations in GABAergic neurons. Prior work indicates that UNC-3 transcriptional regulation of MADD-4/Punctin is essential for proper development of cholinergic synapses with muscles (*Kratsios et al., 2015*). However, *madd-4* is dispensable for cholinergic receptor clustering in neighboring GABAergic neurons (*Barbagallo et al., 2017*) (this study). Thus, cholinergic connectivity with distinct synaptic targets–muscles and GABAergic neurons–is coordinately regulated with cholinergic neuronal identity by *unc-3* transcriptional control of alternate synaptic organizers.

Our work here defines a novel yet essential role for NRX-1 during synapse formation. Neurexin acts in a target-specific manner to coordinate postsynaptic development. Presynaptic neurexin instructs receptor localization and the development of spine-like processes in postsynaptic GABAergic neurons, and *nrx-1* expression is critical for cholinergic transmission onto GABAergic neurons, while not required for signaling onto neighboring muscle. Our work suggests synaptic target-specific utilization of organizers such as neurexin may specify divergent connectivity, and provide a molecular mechanism for target-specific regulation of synapse development.

# Materials and methods

## Strains

*C. elegans* strains were maintained at room temperature (22–24°C) on nematode growth media plates (NGM) seeded with the *Escherichia coli* strain OP50. All strains are derivatives of the N2 Bristol strain (wild type). Transgenic strains were obtained by microinjection to achieve germline transformation (*Mello et al., 1991*) and identified with co-injection markers as previously (*Barbagallo et al., 2017*). Integrated lines were produced by X-ray irradiation and outcrossed to wild type. A complete list of all strains used in this work is included in *Supplementary file 1*.

## Molecular biology

Plasmids were constructed using the two-slot Gateway Cloning system (Invitrogen) as described previously (*Bhattacharya et al., 2014*) and confirmed by restriction digest and/or sequencing as appropriate. All plasmids and primers used in the study are described in *Supplementary files 2* and *3* respectively.

*Tagged receptor constructs:* To generate *flp-13*::ACR-12::GFP::3xHA, ACR-12::GFP was amplified from pDEST-38, removing the original stop codon and adding a 3X HA tag. The product (3195 bp) was ligated into a destination vector to generate pDEST-113. pDEST-113 was recombined with pENTR-3'-*flp-13* to create pAP138 (*flp-13*::ACR-12::GFP::3xHA). To generate *flp-13*::UNC-63::GFP, 5' and 3' fragments of *unc-63* cDNA were PCR amplified from pDEST-57, ligated into pPD117.01 (*mec-7*::GFP), converted into the destination vector pDEST-79, and recombined with pENTR-5'-*flp-13* to create pAP84 (*flp-13*::UNC-63::GFP), where GFP is inserted into the intracellular loop of UNC-63.

*AChR subunit and accessory rescue constructs:* Wild type *unc-38* (1536 bp), *unc-63* (1509 bp), *unc-74* (1344 bp), *unc-50* (906 bp), and *ric-3* (1137 bp) rescue constructs were PCR amplified and ligated into destination vectors to generate pDEST-51, pDEST-57, pDEST-58, pDEST-56, and pDEST-59, respectively. Each of these was recombined with pENTR-*unc-47* to create pAP45 (*unc-47*::*unc-38* cDNA), pAP59 (*unc-47*::*unc-63* cDNA), pAP53 (*unc-47*::*unc-74* cDNA), pAP57 (*unc-47*::*unc-50* cDNA), and pAP55 (*unc-47*::*ric-3* cDNA).

*nrx-1 reporter and rescue constructs:* To generate the 5 kb *nrx-1$_L$*::GFP transcriptional reporter, the *nrx-1$_L$* promoter was amplified from wild type genomic DNA (−4786 bp relative to start) and cloned into pENTR-D-TOPO to generate pENTR-5'-*nrx-1$_L$*. pENTR-5'-*nrx-1$_L$* was then recombined with pDEST-93 (GFP) to generate pAP156 (*nrx-1$_L$*::GFP). The 2 kb *nrx-1$_L$* promoter::GFP fusion construct (−2033 bp relative to start) was created using the same strategy, recombining pENTR-5'-*nrx1$_L$2kb* with pDEST-93 to create pAP118 (*nrx-1$_L$2kb*::GFP). *nrx-1$_L$COEΔ*::GFP (pAP178) was

generated by PCR amplification using mutant primers that disrupt the COE motif (TCCCAAAGGG >TAAAAAAGGG).

Rescuing NRX-1$_L$ minigene constructs were generated by ligation of a 10,598 bp NheI fragment of the *nrx-1* genomic locus extending from the *nrx-1$_L$* start to exon 21 (amplified from cosmid C29A12) with a 715 bp fragment of the *nrx-1* cDNA (isoform A) amplified from a plasmid containing the *nrx-1* coding sequence (GM470, provided by Kang Shen [**Maro et al., 2015**]), and converted into the destination vector pDEST-143. For cell-specific rescue, pDEST-143 was recombined with pENTR-3'-*unc17β*, pENTR-3'-*unc47*, pENTR-3'-*myo3*, and pENTR-3'-*unc3* to create minigene plasmids pAP204 (*unc-17β::nrx-1$_L$*), pAP206 (*unc-47::nrx-1$_L$*), pAP208 (*myo-3::nrx-1$_L$*), and pAP202 (*unc-3::nrx-1$_L$*), respectively. A *nrx-1$_L$* gene fragment (Integrated DNA Technologies) lacking sequence encoding the PDZ binding domain was synthesized, digested and ligated into pAP204 (*unc-17β::nrx-1$_L$*) to generate the pCL83 rescuing construct (*unc-17β::nrx-1$_L$ΔPDZ*).

*unc-129*::NRX-1$_L$::GFP was generated by conversion of the NRX-1$_L$::GFP plasmid GM477 (provided by Kang Shen [**Maro et al., 2015**]) into the destination vector pDEST-99 and recombination with pENTR-3'-*unc129* to create pAP120 (*unc-129*::NRX-1$_L$::GFP). *unc-129*::NRX-1$_L$ΔPDZ::GFP was generated by amplifying *nrx-1$_L$* (from start to PDZ binding domain) and GFP from pAP120 and ligating into a construct containing *unc-129* promoter to generate pAP199 (*unc-129*::NRX-1$_L$ΔPDZ::GFP).

*Chrimson and GCaMP constructs:* To generate *acr-2*::Chrimson, Chrimson coding sequence was amplified from *odr-7*::Chrimson (construct provided by Dirk Albrecht [**Larsch et al., 2015**]) and ligated into a destination vector to create pDEST-104. pDEST-104 was recombined with pENTR-5'-*acr2* to create pRB2 (*acr-2*::Chrimson). To generate *ttr-39*::GCaMP6s::SL2::mCherry, GCaMP6s was amplified from pGP-CMV-GCaMP6s (Addgene) and ligated into a destination vector to create pDEST-95. pDEST-95 was recombined with pENTR-5'-*ttr39* (promoter provided by David Miller [**Petersen et al., 2011**]) to create pAP130 (*ttr-39*::GCaMP6s::SL2::mCherry).

## Confocal microscopy

For all imaging, nematodes were immobilized with sodium azide (0.3 M) on a 2 or 5% agarose pad. Each n represents analysis of the nerve cord from an independent animal. Images were obtained using either a 3i (Intelligent Imaging Innovations) Everest spinning-disk confocal microscope or Olympus BX51WI spinning disk confocal equipped with a 63x objective. All DD1 confocal images were obtained by imaging L4 hermaphrodites of similar size in the region near the pharynx using identical image and laser settings for each marker, and receptor clusters were quantified in a region from the DD1 cell body to connecting commissure (i.e. the synaptic region).

Analysis of synapse number/fluorescence intensity was conducted using either Volocity 6.3 or ImageJ software (open source) using defined intensity threshold values acquired from control experiments for each fluorescent marker. Specifically, the 'find objects' function in Volocity was used, excluding objects > 10 μm$^2$ and <0.2 μm$^2$. Alternatively, the 'analyze particles' function of ImageJ was used. For ImageJ analyses, background fluorescence was first subtracted by calculating the average intensity of each image in a region devoid of puncta. In some cases, fluorescence intensity within a region of interest was also measured and normalized to wild type control as indicated. Confocal montages of the nerve cord were assembled by using the 'straighten to line' function in ImageJ. Only images where the DD1 neuron was clearly distinguishable from neighboring cells were included in analyses.

For measurements of dendritic morphology, post-synaptic protrusions in the ventral dendritic region anterior to the DD1 soma (i.e. the synaptic region) were quantified in L4 hermaphrodites. For posterior DD neurons, post-synaptic protrusions were quantified anterior to the DD3 soma (25 μm region). All protrusions $\geq$ 0.3 μm were analyzed, measuring from the base of the main dendritic process to the tip of the protrusion.

For imaging and quantification of *flp-13*::ACR-12::GFP in **Figure 3A** and **Figure 2—figure supplement 1B**, strains IZ1458 (*ufIs126*) and IZ1557 (*ufIs126; acr-12(ok367)*) were included in the analysis as wild type control. There was no appreciable difference in the number of receptor clusters between the two strains (*ufIs126; acr-12(ok367)* 14.4 ± 0.6, n = 49; *ufIs126* 14.8 ± 0.6, n = 48). ACR-12 receptor clusters positioned within 0.5 μm of the dendritic shaft were scored as dendritic, while receptors outside this region were scored as associated with spiny protrusions.

## Staging and timecourse of receptors/protrusion growth during development

Spine and receptor cluster number in the DD1 synaptic region were analyzed in synchronized animals using strains IZ1458 and IZ1464 as described previously (He et al., 2015). Briefly, embryos for each strain were picked to separate 60 mm unseeded plates and allowed to hatch for 40 min. Newly hatched L1 larvae were moved to freshly seeded plates, and the midpoint of the 40 min in which the embryos hatched was considered t = 0. Plates were incubated at 25°C for 28, 34, 46, and 52 hr. Developing protrusions $\geq$ 0.2 µm in the synaptic region were analyzed, and ACR-12 receptor clusters were quantified as above.

## Single molecule RNA fluorescent in situ hybridization (FISH) and imaging

Custom Stellaris FISH probes against *nrx-1* mRNA were obtained from Biosearch Technologies as a mix of 48 probes conjugated to CAL Fluor Red 590 Dye. Experiments were performed using wild type, *unc-3(e151)* mutants, and *nrx-1(nu485)* mutants expressing either *unc-47*::GFP (*oxIs12*), *unc-4*:: GFP (*wdIs5*), or *unc-17::GFP* (*vsIs48*) markers to label populations of GABA and ACh motor neurons. Synchronized populations of L3-L4 larval animals were fixed and hybridized as described previously (*Ji and van Oudenaarden, 2012*; *Raj et al., 2008*). Images were obtained using spinning disk microscopy as above. Z-projections were analyzed in ImageJ using the 'analyze particles' function. Following background subtraction, the total number of *nrx-1* mRNA molecules was calculated for a 45 x 5.5 µm straightened region of the anterior ventral nerve cord using a defined intensity threshold across all images.

## Injection of fluorescent antibodies for in vivo labeling of nAChRs

For staining of ACR-12 receptors at the cell surface, mouse monoclonal α-HA antibodies (16B12) coupled to Alexa594 were diluted in injection buffer (20 mM $K_3PO_4$, 3 mM K citrate, 2% PEG 6000, pH 7.5). Antibody was injected into the pseudocoelom of early L4 stage wild type or *nrx-1(wy778)* animals as described previously (*Gottschalk and Schafer, 2006*). Animals were allowed to recover for six hours on seeded NGM plates. Only animals in which fluorescence was observed in coelomocytes (indicating uptake of excess antibody and successful injection) were included in the analysis. Injections of anti-GFP Alexa594 antibody followed the same protocol.

## *nrx-1* behavioral assays

One-day old adults were placed on thinly seeded NGM plates and tracked for a period of 5 min using Single Worm Tracker 2.0 (WT2) (*Yemini et al., 2011*). Worm tracker software version 2.0.3.1, created by Eviatar Yemini and Tadas Jucikas (Schafer lab, MRC, Cambridge, UK), was used to analyze movement.

## Calcium imaging

Transgenic animals expressing *ttr-39*::GCaMP6s::SL2::mCherry (GABA neurons) or *myo-3*:: NLSwCherry::SL2::GCaMP6s (muscle, from M. Alkema) along with *acr-2*::Chrimson (cholinergic neurons) were placed on plates seeded with OP50 containing 2.75 mM All-Trans Retinal (ATR) for 24 hr prior to experiments. Young adults were immobilized on 5% agarose pads in 2,3-Butanedione monoxime (BDM) (30 mg/ml). For all genotypes, control animals grown in the absence of ATR were imaged.

Imaging was carried out on a Yokogawa CSU-X1-A1N spinning disk confocal system equipped with EM-CCD camera (Hammamatsu, C9100-50) and 40X C-Apochromat 1.2 NA water immersion objective. Optogenetic stimulation experiments employed a 625 nm (40 W) LED (Mightex Systems). Optical output through the objective was 0.3 mW/mm² at the focal plane of the specimen. Simultaneous GCaMP excitation (488 nm) and emission (525 nm) acquisition and Chrimson activation were achieved using a 556 nm edge BrightLine single-edge short-pass dichroic beam splitter positioned in the light path (Semrock).

Data were acquired using Volocity software. Images were binned at 4 × 4 during acquisition and sampled at 10 Hz. GABA motor neuron and muscle ROIs in respective experiments were identified by mCherry fluorescence. Recordings from motor neuron cell bodies were obtained systematically,

beginning at the anterior end of the ventral nerve cord and moving in a posterior direction. Each field typically contained 1–5 GABA motor neurons. Only recordings of neurons located anterior to the vulva were included in the analysis. Muscle recordings were obtained either directly anterior or posterior to the vulva.

Photobleaching correction was carried out by fitting an exponential function to the data (Correct-Bleach plugin, ImageJ). A linear fit (Igor Pro, Wavemetrics) of the background fluorescence was subtracted from the cell body fluorescence across all time points. Pre-stimulus baseline fluorescence ($F_0$) was calculated as the average of the corrected background-subtracted data points in the first 4 s of the recording and the corrected fluorescence data was normalized to prestimulus baseline as $\Delta F/F_0$, where $\Delta F = F - F_0$. Peak $\Delta F/F_0$ was determined by fitting a Gaussian function to the $\Delta F/F_0$ time sequence using Multi peak 2.0 (Igor Pro, WaveMetrics). All data collected were analyzed, including failures (no response to stimulation). Peak $\Delta F/F_0$ values were calculated from recordings of >10 animals per genotype. Mean peaks ± SEM were calculated from all peak $\Delta F/F_0$ data values and normalized to the wild type mean. Latency was calculated as the time required from stimulus onset for fluorescence ($\Delta F/F_0$) to reach two times the pre-stimulus baseline standard deviation (*Larsch et al., 2015*). Duration was measured as the time between the onset of the transient and the completion of the decay back to the baseline. Rise and decay time constants were determined from the time constant of exponential fits between the baseline and peak fluorescence as appropriate.

## Acknowledgements

We would like to thank Dori Schafer, Vivian Budnik, Alexandra Byrne and members of the Francis lab for critical reading of the manuscript, Claire Bénard, Kang Shen, and David Miller for sharing reagents, and Michael Gorczyca, Will Joyce, and the UMMS Model Organism CRISPR Core for technical assistance. Some nematode strains used in this work were provided by the *Caenorhabditis* Genetics Center, which is funded by the NIH National Center for Research Resources (NCRR). This research was supported by NIH NINDS R01 NS064263 (MMF), NIH NIDA F31 DA038399 (AP), and NIH NIGMS R01 GM084491 (MJA).

## Additional information

### Funding

| Funder | Grant reference number | Author |
| --- | --- | --- |
| National Institute of Neurological Disorders and Stroke | R01NS064263 | Michael M Francis |
| National Institute on Drug Abuse | F31DA038399 | Alison Philbrook |
| National Institute of General Medical Sciences | R01GM084491 | Mark J Alkema |

The funders had no role in study design, data collection and interpretation, or the decision to submit the work for publication.

### Author contributions

Alison Philbrook, Conceptualization, Formal analysis, Funding acquisition, Validation, Investigation, Methodology, Writing—original draft, Writing—review and editing, AP, Wrote the paper, Assisted with experimental design, Carried out all aspects of experiments, Collected and analyzed the data, Assisted with preparing the manuscript, Assisted with funding support; Shankar Ramachandran, Conceptualization, Formal analysis, Validation, Investigation, Methodology, Writing—review and editing, SR, Designed the calcium imaging experiments, Carried out all aspects of the calcium imaging experiments and collected the data, Assisted with preparing the manuscript; Christopher M Lambert, Investigation, Methodology, CL, Assisted with the generation of all molecular biology reagents and strains; Devyn Oliver, Formal analysis, Investigation, Writing—review and editing, DO, Assisted with strain generation and fluorescent microscopy experiments, data collection and analysis, Assisted with manuscript preparation; Jeremy Florman, Formal analysis, Investigation, JF, Assisted

with behavioral experiments, data collection and analysis; Mark J Alkema, Supervision, Funding acquisition, MJA, Assisted with funding for the project and supervision of behavioral experiments; Michele Lemons, Conceptualization, Investigation, Methodology, Writing—review and editing, ML, Conducted FISH experiments, data collection and analysis, Assisted with manuscript preparation; Michael M Francis, Conceptualization, Resources, Formal analysis, Supervision, Funding acquisition, Validation, Methodology, Writing—original draft, Writing—review and editing, MMF, Supervised all experiments, Assisted with experimental design and analysis, Wrote the paper, Provided funding support

### Author ORCIDs
Alison Philbrook (ID) http://orcid.org/0000-0003-3330-3086
Jeremy Florman (ID) http://orcid.org/0000-0001-7578-3511
Michael M Francis (ID) http://orcid.org/0000-0002-8076-6668

### Decision letter and Author response
Decision letter https://doi.org/10.7554/eLife.35692.041
Author response https://doi.org/10.7554/eLife.35692.042

## Additional files

### Supplementary files
• Supplementary file 1. Strain list.
DOI: https://doi.org/10.7554/eLife.35692.036

• Supplementary file 2. Plasmid list.
DOI: https://doi.org/10.7554/eLife.35692.037

• Supplementary file 3. Primer list.
DOI: https://doi.org/10.7554/eLife.35692.038

• Transparent reporting form
DOI: https://doi.org/10.7554/eLife.35692.039

### Data availability
All data generated or analysed during this study are included in the manuscript and supporting files.

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
