## [Decision Letter]

Thank you for submitting your article "Neurexin directs partner-specific synaptic connectivity in *C. elegans*" for consideration by *eLife*. Your article has been reviewed by three peer reviewers, and the evaluation has been overseen by a Reviewing Editor (Oliver Hobert) and Marianne Bronner as the Senior Editor. The reviewers have opted to remain anonymous.

As you will see below, the overall importance of the manuscript was appreciated by the reviewers, but each of the reviewers identified a number of points that need to be addressed. Some of these points require experimentation, while many others can be addressed through editorial changes.

Reviewer #1:

This paper features a genetically regulated pathway that functions in presynaptic cholinergic neurons to control morphogenesis and receptor clustering at post synaptic domains in a mechanism that depends on the conserved cell surface protein, Neurexin. Interestingly, the function of this synaptogenic pathway is limited to only one of the two types of postsynaptic cells receiving cholinergic innervation. The authors emphasize the selectivity of this effect on "divergent" synapses which will no doubt stimulate future studies to reveal the cell biological mechanism of this example of synaptic discrimination. This work exploits the power of live cell imaging and genetics in *C. elegans* to good effect. Experiments are comprehensive, and the paper is well written. As noted below, however, additional experiments are needed to validate key findings.

1) CRISPR-induced mutations that are predicted to disrupt translation of the C-terminal PDZ domain are used in structure-function experiments to argue that the PDZ domain is required for NRX-1 function. This approach does not rule out the alternative explanation, however, that expression of the entire NRX-1 protein is also disrupted. Indeed, this possibility seems very likely for the nrx-1(uf181) allele which introduces a premature stop codon and thus is predicted to result in nonsense mediated decay (NMD) of the entire nrx-1 transcript, and hence, a mutant phenotype equivalent to that of nrx-1 null alleles. To remedy this ambiguity, the authors need to show either that the truncated NRX-1 protein is expressed in the CRISPR alleles or demonstrate that a NRX-1 protein lacking the PDZ domain fails to rescue the nrx-1 mutant.

2) The nrx-1 mutation results in fewer ACR-12 clusters which the authors interpret to mean that NRX-1 is required for ACR-12 clustering. This result could also arise, however, from reduced acr-12 expression. This alternative possibility should be addressed.

3) Are spine-like protrusions unique to the DD1 neuron or are they also visible in other members of the DD class? This point seems important since ACR-12 receptor clustering is disrupted in other DD neurons.

4) In subsection “The synaptic organizer neurexin directs post-synaptic development in a partner-specific manner”. The authors state "…we find that nrx-1 is required for ACR-12 receptor localization in GABAergic DD and VD neurons, although we observe spine-like protrusions only in DD neurons. These findings argue that NRX-1 is not solely involved in directing spine development but serves an additional role in receptor clustering." Did this work rule out the existence of post synaptic spines in VDs or was it simply not possible to resolve them due to overlapping GFP signal from DDs in the dorsal cord? This point needs to be addressed.

Reviewer #2:

Philbrook et al., examine the role of the cell adhesion molecule neurexin in synapse assembly, and find that *C. elegans* presynaptic neurexin drives the development of one but not the other postsynaptic partner in a dyadic synapse. They show that nrx is responsible for the development of dendritic "spines" on postsynaptic GABAergic neurons as well as the clustering/stabilization of acetylcholine receptors opposite sites of presynaptic release. Loss of nrx leads to a loss of spines, AChR clusters, and functional connectivity between these two classes of neurons, while leaving intact the development and connectivity onto muscle cells. The authors go on to demonstrate the transcriptional control over nrx expression in these neurons. This is a technically impressive study that lends some insight into how individual neurons might form molecularly distinct synapses onto multiple partners, and perhaps also identifies a new model system for studying dendritic "spine" development.

The weakness of the paper is that the molecular conclusions are not novel (neurexin is well known to be involved in trans-synaptic receptor clustering). In addition, the effect of neurexin in regulating postsynaptic development could be indirect, possibly due to the presynaptic deficits. Lastly, the main conceptual conclusions regarding dyadic synapse development and/or the establishment of a model for spine development in the worm are not adequately fleshed out.

1) The authors fail to show that the effect of nrx is mediated by a direct interaction with a postsynaptic partner, as opposed to through disruption of presynaptic assembly. This is especially salient given the lack of phenotype in neuroligin mutants or any other putative postsynaptic partner. Their evidence for a direct role in postsynaptic development relies on the fact that they can disrupt nrx function with mutations in the extracellular domain, but these could still have an effect on the protein as a whole. In fact, in their candidate screen they found that syd-1 mutants, which are known to play a role in presynaptic assembly and to bind nrx in other organisms, also exhibit a defect in postsynaptic receptor clustering. The authors should determine whether other (stronger) presynaptic assembly mutants such as syd-2/liprin-α also exhibit defects in postsynaptic receptor clustering. If so, this could indicate a more indirect role for nrx. Their analysis of presynaptic development is cursory (they look at one synaptic vesicle marker expressed in all cholinergic neurons), so their conclusions that presynaptic development is normal should be much more tempered.

2) A major conclusion of the paper is that neurexin is responsible for the proper development and function of one half of a dyadic synapse. This is a novel and intriguing conclusion, but the idea that these specific synapses are functionally dyadic rests primarily on the EM connectomics data. The functional assays performed here were not performed at the same synapses that were developmentally characterized in the first part of the paper: according to the methods section, muscle function was assessed near the vulva, and GABAergic function was assessed in many different GABAergic neurons. Moreover, presynaptic stimulation was targeted quite broadly to all cholinergic neurons. To properly conclude that the DD1 synapses, whose development is dependent on nrx, are functionally part of a dyadic synapse, the authors should record calcium transients from the muscles that are in the same region as DD1, and from DD1 itself rather than all GABAergic neurons. Moreover, to exclude the possibility that different classes of presynaptic neurons exert preferentially control of different postsynaptic partners, they should express their Chrimson in as small a subset of presynaptic neurons as possible (for example, VAs, for which there is a specific promoter, unc-4). Showing that a functional readout from the same location is differentially controlled by nrx would greatly strengthen their main conclusion.

3) Another interesting facet of the current study is the possibility to establish this as a system in which to study dendritic spine development in such a genetically tractable organism. If the authors choose to emphasize this aspect of the work though, they should demonstrate that these structures indeed behave like spines, for example, by showing that there is some intracellular signaling component that is restricted to the spines. Perhaps more importantly, it is unclear how neurexin acts non-cell-autonomously to promote initial spine outgrowth, which the authors suggest, due to the lack of spines even early in development. Given the significant gap between the pre- and postsynaptic neurons, it seems unlikely that presynaptic nrx could directly induce the initial outgrowth of these spines. More likely, if the interaction is indeed direct, nrx stabilizes spines that have already grown out. To determine whether this is the case, and strengthen their model of spine development, the authors could conduct live imaging of the spines during their development, to see whether initial spine outgrowth is coupled with retraction.

Reviewer #3:

In this study the authors make use of cell biology, molecular genetics, behavior and calcium imaging to determine the mechanisms that specify formation of postsynaptic sites in GABA neurons. They discover that transcription factor unc-3 directs expression in presynaptic cholinergic cells of Neurexin, and that Neurexin acts non-cell autonomously to specify both the formation of dendritic spine-like structures, and clustering of cholinergic postsynaptic receptors. Their studies also demonstrate that the molecular mechanisms used by Neurexin to instruct formation of postsynaptic sites are independent of the canonical interacting partner, Neuroligin. This suggests that the newly identified, in vivo mechanisms are different from those that have been previously reported. Importantly, the mechanisms used by cholinergic neurons to connect to GABAergic neurons are distinct from the mechanisms used by the same cholinergic neurons to form neuromuscular junctions. The study is elegant, rigorous and a significant contribution to the field.

1) This work has a number of interesting contributions, including the one that the authors chose to emphasize, the concept of genetically separable molecular mechanism specifying partner choice in dyadic synapses (which they refer to as divergent synapses, a term that should be defined early in the Introduction). If the framing remains on dyadic synapses, the paper would benefit from the authors reproducing and showing the muscle phenotypes they refer to throughout the text.

2) Have the authors validated ACR-12:GFP construct as functional with phenotypic rescue in acr-12 mutants?

3) In Figure 1F, the authors observed 60%-70% of the cholinergic receptor clusters are located on the spine-like structure and 30-40% are located on the main dendrites. Do nrx-1 mutants reduce ACR-12 clustering to the same level in both pools?

4) In VD GABAergic neurons there are no spiny protrusions, but the ACR-12 clustering is still reduced in nrx-1 mutants. Could the authors comment on these differences between DD neurons and VD neurons (in terms of the spine-like structures)?

5) Mutations in the acetylcholine receptor subunit genes, such as unc-63 and acr-12, reduce cholinergic receptor clustering without affecting spiny protrusions. Are there any mutants that abolish spiny protrusion without affecting receptor clustering in the DD neurons?

6) In Figure 3E, what does the allele tm1961 do to nrx-1? Does it not cause decrease in ACR-12 cluster numbers?

7) In Figure 2D, UNC-29::GFP or UNC-63::GFP were expressed in DD neurons and their pattern resembles ACR-12::GFP clustering. Were the three subunits co-expressed with different fluorophores? Was their relative localization to each other (or co-localization) examined?

8) In Figure 6G (left), it seems there is a significant difference between wild-type and unc-3(e151)+Ach:nrx-1L. If so, does lacking NRX-1L alone explain the reduced cluster number in unc-3(e151)?

9) Was unc-3 rescued in cholinergic neurons?

10) The authors demonstrate that the intracellular domain of Neurexin is required for its function, yet Neurexin plays a non-cell autonomous role in postsynaptic vesicle clustering. Can the authors integrate their observations of the intracellular domain requirement to the model/discussion?

---

## [Author Response]

Reviewer #1:This paper features a genetically regulated pathway that functions in presynaptic cholinergic neurons to control morphogenesis and receptor clustering at post synaptic domains in a mechanism that depends on the conserved cell surface protein, Neurexin. Interestingly, the function of this synaptogenic pathway is limited to only one of the two types of postsynaptic cells receiving cholinergic innervation. The authors emphasize the selectivity of this effect on "divergent" synapses which will no doubt stimulate future studies to reveal the cell biological mechanism of this example of synaptic discrimination. This work exploits the power of live cell imaging and genetics in C. elegans to good effect. Experiments are comprehensive, and the paper is well written. As noted below, however, additional experiments are needed to validate key findings.

We thank the reviewer for their positive comments.

1) CRISPR-induced mutations that are predicted to disrupt translation of the C-terminal PDZ domain are used in structure-function experiments to argue that the PDZ domain is required for NRX-1 function. This approach does not rule out the alternative explanation, however, that expression of the entire NRX-1 protein is also disrupted. Indeed, this possibility seems very likely for the nrx-1(uf181) allele which introduces a premature stop codon and thus is predicted to result in nonsense mediated decay (NMD) of the entire nrx-1 transcript, and hence, a mutant phenotype equivalent to that of nrx-1 null alleles. To remedy this ambiguity, the authors need to show either that the truncated NRX-1 protein is expressed in the CRISPR alleles or demonstrate that a NRX-1 protein lacking the PDZ domain fails to rescue the nrx-1 mutant.

We thank the reviewer for raising this important point. We agree that the effects of the CRISPR alleles may not be restricted to the PDZ binding motif, and this may complicate interpretation of our findings using these mutants. We have therefore removed these alleles from the revised manuscript and instead, as the reviewer suggests, show that cholinergic expression of a *nrx-1_L_* rescuing construct lacking the PDZ binding domain rescues ACR-12::GFP clustering defects in *nrx-1* mutants (Figure 5—figure supplement 2A-B). Further, we show that a *nrx-1_L_* transgene lacking the predicted PDZ binding sequence localizes to cholinergic synaptic terminals, suggesting that presynaptic localization can occur without PDZ protein binding to this motif (Figure 5—figure supplement 2C-E). These results argue that protein interactions via the PDZ binding motif may not be required for directing cholinergic connectivity with GABAergic neurons and are discussed in subsection “*nrx-1* is expressed and functionally required in cholinergic motor neurons” of the revised submission.

2) The nrx-1 mutation results in fewer ACR-12 clusters which the authors interpret to mean that NRX-1 is required for ACR-12 clustering. This result could also arise, however, from reduced acr-12 expression. This alternative possibility should be addressed.

To address this possibility, we quantified the fluorescence of a *flp-13*::mCherry transcriptional reporter (the same promoter used for cell-specific expression of ACR-12::GFP) in the DD1 neuron of *nrx-1* deletion animals compared with wild type. We did not observe a significant reduction in *flp-13*::mCherry fluorescence intensity (*p*=0.52, student’s t-test), arguing against reduced transgene expression in *nrx-1* mutants. We now clarify this point in Figure 3—figure supplement 2C-D and in subsection “Neurexin directs cholinergic connectivity with GABAergic neurons” of the text.

3) Are spine-like protrusions unique to the DD1 neuron or are they also visible in other members of the DD class? This point seems important since ACR-12 receptor clustering is disrupted in other DD neurons.

This is an interesting point. We now show that spine-like protrusions are visible in other DD neurons (Figure 1—figure supplement 1C), include quantification in Figure 4C, and discuss this result in subsection “Clusters of the GFP-tagged acetylcholine receptor subunit ACR-12 are localized to spine-like dendritic protrusions on the DD1 GABAergic neuron” and subsection “Post-synaptic morphological development requires NRX-1” of the revised text. Mutation of *nrx-1* disrupts spine-like protrusions in both DD1 and more posterior DD neurons.

4) In subsection “The synaptic organizer neurexin directs post-synaptic development in a partner-specific manner”. The authors state "…we find that nrx-1 is required for ACR-12 receptor localization in GABAergic DD and VD neurons, although we observe spine-like protrusions only in DD neurons. These findings argue that NRX-1 is not solely involved in directing spine development but serves an additional role in receptor clustering." Did this work rule out the existence of post synaptic spines in VDs or was it simply not possible to resolve them due to overlapping GFP signal from DDs in the dorsal cord? This point needs to be addressed.

Using a reporter that labels VD and DD processes, we did not find postsynaptic spines in VD neurons (representative image now included in Figure 1—figure supplement 1D). However, as the reviewer notes, we cannot rule out the possibility that these features may be obscured by the density of nerve cord processes. This point is now clarified in subsection “Clusters of the GFP-tagged acetylcholine receptor subunit ACR-12 are localized to spine-like dendritic protrusions on the DD1 GABAergic neuron” of the revised text.

Reviewer #2:Philbrook et al., examine the role of the cell adhesion molecule neurexin in synapse assembly, and find that C. elegans presynaptic neurexin drives the development of one but not the other postsynaptic partner in a dyadic synapse. They show that nrx is responsible for the development of dendritic "spines" on postsynaptic GABAergic neurons as well as the clustering/stabilization of acetylcholine receptors opposite sites of presynaptic release. Loss of nrx leads to a loss of spines, AChR clusters, and functional connectivity between these two classes of neurons, while leaving intact the development and connectivity onto muscle cells. The authors go on to demonstrate the transcriptional control over nrx expression in these neurons. This is a technically impressive study that lends some insight into how individual neurons might form molecularly distinct synapses onto multiple partners, and perhaps also identifies a new model system for studying dendritic "spine" development.

We thank the reviewer for their positive assessment.

The weakness of the paper is that the molecular conclusions are not novel (neurexin is well known to be involved in trans-synaptic receptor clustering). In addition, the effect of neurexin in regulating postsynaptic development could be indirect, possibly due to the presynaptic deficits. Lastly, the main conceptual conclusions regarding dyadic synapse development and/or the establishment of a model for spine development in the worm are not adequately fleshed out.

We reply to the reviewer’s specific points below and hope the reviewer will agree that these additional analyses address their concerns.

1) The authors fail to show that the effect of nrx is mediated by a direct interaction with a postsynaptic partner, as opposed to through disruption of presynaptic assembly. This is especially salient given the lack of phenotype in neuroligin mutants or any other putative postsynaptic partner. Their evidence for a direct role in postsynaptic development relies on the fact that they can disrupt nrx function with mutations in the extracellular domain, but these could still have an effect on the protein as a whole. In fact, in their candidate screen they found that syd-1 mutants, which are known to play a role in presynaptic assembly and to bind nrx in other organisms, also exhibit a defect in postsynaptic receptor clustering. The authors should determine whether other (stronger) presynaptic assembly mutants such as syd-2/liprin-α also exhibit defects in postsynaptic receptor clustering. If so, this could indicate a more indirect role for nrx.

We thank the reviewer for their insights and agree that this is an important point to address. We now include new analysis of a syd-2 deletion allele in the revised version of the manuscript. The effects of syd-2 deletion on ACR-12 receptor clustering are substantially less severe than observed for mutation of nrx-1, and comparable to mutation of syd-1 (Figure 3A). As the reviewer notes, our findings related to syd-1 and syd-2 may point to an interesting link between presynaptic organization and postsynaptic assembly, and we plan to pursue these questions in future work. Based on the severity of the nrx-1 deletion phenotype compared with mutation of syd-1 or syd-2, and the specific alterations in GABAergic post-synaptic specializations that we observe (muscle receptors are normally clustered in nrx-1 mutants), we propose a specific requirement for neurexin in guiding post-synaptic assembly in GABA neurons independent of contributions to presynaptic organization.

Their analysis of presynaptic development is cursory (they look at one synaptic vesicle marker expressed in all cholinergic neurons), so their conclusions that presynaptic development is normal should be much more tempered.

The reviewer raises a good point. Our analysis of the SNB-1::GFP synaptic vesicle marker in cholinergic neurons suggests that synaptic vesicles are grossly normally in *nrx-1* mutants, but does not exclude all potential defects in presynaptic development. Based on our observation that evoked stimulation of cholinergic motor neurons produces normal muscle Ca^2+^ transients in *nrx-1* mutants, we suggest that presynaptic organization is sufficiently preserved to support evoked cholinergic transmission, consistent with prior electrophysiology studies of *nrx-1* mutants from the Kaplan lab. We now clarify our discussion of these findings as suggested by the reviewer (subsection “Neurexin directs cholinergic connectivity with GABAergic neurons”).

2) A major conclusion of the paper is that neurexin is responsible for the proper development and function of one half of a dyadic synapse. This is a novel and intriguing conclusion, but the idea that these specific synapses are functionally dyadic rests primarily on the EM connectomics data. The functional assays performed here were not performed at the same synapses that were developmentally characterized in the first part of the paper: according to the methods section, muscle function was assessed near the vulva, and GABAergic function was assessed in many different GABAergic neurons. Moreover, presynaptic stimulation was targeted quite broadly to all cholinergic neurons. To properly conclude that the DD1 synapses, whose development is dependent on nrx, are functionally part of a dyadic synapse, the authors should record calcium transients from the muscles that are in the same region as DD1, and from DD1 itself rather than all GABAergic neurons.

We thank the reviewer for these comments. We now realize that we did not provide a clear explanation of the rationale underlying our Ca^2+^ imaging studies. Our initial light microscopy studies focused on DD1 as an experimentally tractable model, but we note that our findings for DD1 are generalizable to other GABAergic neurons. In the revised version we include new data to make this point more clearly. In particular, we show that more posterior DD neurons also have spines, and that both spine outgrowth and receptor clustering in these neurons require *nrx-1* (Figure 4C and Figure 4—figure supplement 1A). Similarly, we show that acetylcholine receptor clusters in VD neurons are disrupted by mutation of *nrx-1* (Figure 3E-G). We have recorded evoked Ca^2+^ transients directly from DD1 neurons, but, due to the close proximity of DD1 to the pharynx, these recordings are challenging. We therefore included GABAergic neurons posterior to DD1 in our analysis. We clarify our rationale in subsection “*nrx-1* deletion impairs cholinergic neurotransmission onto GABAergic neurons” of the revised text.

Moreover, to exclude the possibility that different classes of presynaptic neurons exert preferentially control of different postsynaptic partners, they should express their Chrimson in as small a subset of presynaptic neurons as possible (for example, VAs, for which there is a specific promoter, unc-4). Showing that a functional readout from the same location is differentially controlled by nrx would greatly strengthen their main conclusion.

For our studies, we elected for broad presynaptic cholinergic stimulation in order to minimize any potential difficulties arising from preferential connectivity. The available connectivity data suggest that individual GABAergic neurons receive inputs from multiple classes of cholinergic neurons. Thus, we reasoned that stimulation of all presynaptic cholinergic neurons would offer the best opportunity for an unbiased investigation of post-synaptic responses. Nonetheless, we agree with the reviewer that the question of preferential connectivity is intriguing. As the reviewer suggests, we generated strains expressing *unc-4*::Chrimson, but, in our initial attempts, have been unable to record Ca^2+^ transients from either GABAergic neurons or muscles following light stimulation. This may be due to low expression of the *unc-4*::Chrimson transgene in A motor neurons, as we observe measurable expression in only a small number of ventral cord neurons (we think these are VC neurons) by the age at which we record (young adult). We hope the reviewer will agree that additional exploration of this interesting question is more suitable for a future study following the optimization of these tools.

3) Another interesting facet of the current study is the possibility to establish this as a system in which to study dendritic spine development in such a genetically tractable organism. If the authors choose to emphasize this aspect of the work though, they should demonstrate that these structures indeed behave like spines, for example, by showing that there is some intracellular signaling component that is restricted to the spines. Perhaps more importantly, it is unclear how neurexin acts non-cell-autonomously to promote initial spine outgrowth, which the authors suggest, due to the lack of spines even early in development. Given the significant gap between the pre- and postsynaptic neurons, it seems unlikely that presynaptic nrx could directly induce the initial outgrowth of these spines. More likely, if the interaction is indeed direct, nrx stabilizes spines that have already grown out. To determine whether this is the case, and strengthen their model of spine development, the authors could conduct live imaging of the spines during their development, to see whether initial spine outgrowth is coupled with retraction.

We agree with the reviewer that our study poses some intriguing questions about dendritic spine development in worms. By L4 stage, we note an appreciable gap between the pre- and postsynaptic domains in *nrx-1* mutants, due to the absence of spine-like protrusions contacting the presynaptic axon (Figure 7A). As the size of this gap is expected to be significantly smaller in younger animals, we envision that NRX-1 promotes spine outgrowth early in development. We have not been able to detect these structures prior to the L2 stage, and we therefore suggest that they form coincident with, or shortly after, DD neuron remodeling, when the post-embryonic ventral cholinergic neurons are integrated into the circuit. To directly address temporal requirements for *nrx-1* expression, we have attempted to delay neurexin expression using an inducible (heat-shock) promoter, but, to date, our attempts have been unsuccessful (due to leaky expression of *nrx-1*). We agree with the reviewer that live imaging of the spines during their outgrowth is likely to be informative, and we are currently pursuing this approach. Nonetheless, these experiments are technically challenging given the small size of these structures in L2 animals and will likely require considerable time for optimization. While we favor the idea that neurexin is required for spine outgrowth, at this point we do not rule out a potential maintenance role. We now include this possibility in the text (subsection “Post-synaptic morphological development requires NRX-1”).

Reviewer #3:In this study the authors make use of cell biology, molecular genetics, behavior and calcium imaging to determine the mechanisms that specify formation of postsynaptic sites in GABA neurons. They discover that transcription factor unc-3 directs expression in presynaptic cholinergic cells of Neurexin, and that Neurexin acts non-cell autonomously to specify both the formation of dendritic spine-like structures, and clustering of cholinergic postsynaptic receptors. Their studies also demonstrate that the molecular mechanisms used by Neurexin to instruct formation of postsynaptic sites are independent of the canonical interacting partner, Neuroligin. This suggests that the newly identified, in vivo mechanisms are different from those that have been previously reported. Importantly, the mechanisms used by cholinergic neurons to connect to GABAergic neurons are distinct from the mechanisms used by the same cholinergic neurons to form neuromuscular junctions. The study is elegant, rigorous and a significant contribution to the field.

We thank the reviewer for their positive assessment.

1) This work has a number of interesting contributions, including the one that the authors chose to emphasize, the concept of genetically separable molecular mechanism specifying partner choice in dyadic synapses (which they refer to as divergent synapses, a term that should be defined early in the Introduction). If the framing remains on dyadic synapses, the paper would benefit from the authors reproducing and showing the muscle phenotypes they refer to throughout the text.

We thank the reviewer for the suggestion. We have more clearly defined divergent connectivity in the Introduction as the reviewer suggested, and have reproduced the muscle phenotypes referred to in the text, now included in Figure 3—figure supplement 2A, B.

2) Have the authors validated ACR-12:GFP construct as functional with phenotypic rescue in acr-12 mutants?

We have previously shown that GABAergic expression of ACR-12::GFP rescues *acr-12* mutant phenotypes, demonstrating that this construct is functional (Petrash et al., 2013). We now indicate this in the text in the Introduction.

3) In Figure 1F, the authors observed 60%-70% of the cholinergic receptor clusters are located on the spine-like structure and 30-40% are located on the main dendrites. Do nrx-1 mutants reduce ACR-12 clustering to the same level in both pools?

In wild type, 67% of receptor clusters are associated with spines, while just 14.7% of receptor clusters are associated with remaining spines in *nrx-1(ok1649)* mutants. In contrast, the number of receptor clusters associated with the main dendritic shaft is not appreciably altered by mutation of *nrx-1 (p*=0.67, student’s t test). This point is now clarified in the text (subsection “Post-synaptic morphological development requires NRX-1”).

4) In VD GABAergic neurons there are no spiny protrusions, but the ACR-12 clustering is still reduced in nrx-1 mutants. Could the authors comment on these differences between DD neurons and VD neurons (in terms of the spine-like structures)?

Using a reporter that labels VD and DD processes, we did not find postsynaptic spines in VD neurons (representative image now included in Figure 1—figure supplement 1D). This suggests either neurexin promotes receptor clustering independently of spines in VD neurons, or spines are present, but we are unable to detect them due to the number of GABAergic processes. Please also see the response to reviewer 1.

5) Mutations in the acetylcholine receptor subunit genes, such as unc-63 and acr-12, reduce cholinergic receptor clustering without affecting spiny protrusions. Are there any mutants that abolish spiny protrusion without affecting receptor clustering in the DD neurons?

Independent of this study, we have isolated several mutants from a forward genetic screen in which spine-like protrusions are significantly decreased, but receptor clustering is less strongly affected, perhaps indicating a redistribution of receptor clusters to the main dendritic shaft. We are currently in the process of identifying and characterizing the genes affected by these mutations.

6) In Figure 3E, what does the allele tm1961 do to nrx-1? Does it not cause decrease in ACR-12 cluster numbers?

*nrx-1(tm1961)* significantly reduces ACR-12 receptor clusters, ***p*<0.01, ANOVA with Dunnett’s multiple comparisons test (updated in Figure 3E).

7) In Figure 2D, UNC-29::GFP or UNC-63::GFP were expressed in DD neurons and their pattern resembles ACR-12::GFP clustering. Were the three subunits co-expressed with different fluorophores? Was their relative localization to each other (or co-localization) examined?

Great idea. We are still working on creating constructs to co-express subunits with different fluorophores. To date, we have not identified additional fluorophores that provide sufficient fluorescent signal for analysis, and also accommodate proper receptor assembly and trafficking. We hope to re-examine the relative co-localization of the three subunits in the future as we develop new reagents to address this question.

8) In Figure 6G (left), it seems there is a significant difference between wild-type and unc-3(e151)+Ach:nrx-1L. If so, does lacking NRX-1L alone explain the reduced cluster number in unc-3(e151)?

Expression of *unc-3*::NRX-1_L_::GFP in *unc-3* mutants does not restore ACR-12::GFP receptor clustering fully to wild type levels. However, in this analysis we are using a different promoter (*unc-3* promoter compared to the *unc-17β* promoter used in Figure 5C-D for rescue) so the levels of expression likely differ across these two strains. We suggest that decreased *nrx-1_L_* expression in *unc-3* mutants is a major contributor to decreased receptor clustering. However, reduced *nrx-1_L_* expression is unlikely to account for all of the defects. For example, mutation of *unc-3* produces variable changes in process outgrowth and nerve cord fasciculation, which do not occur with disruption of *nrx-1*, and therefore may influence our analysis of receptor clustering in *unc-3* mutants and rescue animals. We now clarify this point in the text (subsection “The COE-type transcription factor *unc-3* directly controls neurexin expression”).

9) Was unc-3 rescued in cholinergic neurons?

We have previously demonstrated that *unc-3* is required in cholinergic motor neurons for ACR12 receptor clustering (Barbagallo et al., 2017).

10) The authors demonstrate that the intracellular domain of Neurexin is required for its function, yet Neurexin plays a non-cell autonomous role in postsynaptic vesicle clustering. Can the authors integrate their observations of the intracellular domain requirement to the model/discussion?

We have now revised this model in response to comments from reviewer 1. Specifically, we have removed the CRISPR alleles from the revised manuscript, and instead show that cholinergic expression of a *nrx-1_L_*rescuing construct lacking the PDZ binding domain rescues ACR-12 clustering defects in *nrx-1* mutants. Based on these findings, we propose that NRX-1 can promote post-synaptic assembly in GABAergic neurons without a requirement for protein interactions via the PDZ binding motif. We have revised the text accordingly (subsection “*nrx-1* is expressed and functionally required in cholinergic motor neurons”).